# Evolutionarily stable gene clusters shed light on the common grounds of pathogenicity in the *Acinetobacter calcoaceticus-baumannii* complex

**Bardya Djahanschiri**[1], **Gisela Di Venanzio**[2], **Jesus S. Distel**[2], **Jennifer Breisch**[3], **Marius Alfred Dieckmann**[4], **Alexander Goesmann**[4], **Beate Averhoff**[3], **Stephan Göttig**[5], **Gottfried Wilharm**[6], **Mario F. Feldman**[2], **Ingo Ebersberger**[1,7,8]*

1 Applied Bioinformatics Group, Inst. of Cell Biology and Neuroscience, Goethe University Frankfurt, Frankfurt am Main, Germany, 2 Department of Molecular Microbiology, Washington University School of Medicine, St Louis, Missouri, United States of America, 3 Inst. of Molecular Biosciences, Department of Molecular Microbiology and Bioenergetics, Goethe University Frankfurt, Frankfurt am Main, Germany, 4 Bioinformatics and Systems Biology, Justus Liebig University Gießen, Gießen, Germany, 5 Institute for Medical Microbiology and Infection Control, University Hospital, Goethe University, Frankfurt, Germany, 6 Robert Koch Institute, Project group P2, Wernigerode, Germany, 7 Senckenberg Biodiversity and Climate Research Centre (S-BIKF), Frankfurt am Main, Germany, 8 LOEWE Center for Translational Biodiversity Genomics (TBG), Frankfurt am Main, Germany

* ebersberger@bio.uni-frankfurt.de

**Data Availability Statement:** The authors confirm that all data underlying the findings are fully available without restriction. All numerical data that

## Abstract

Nosocomial pathogens of the *Acinetobacter calcoaceticus-baumannii* (ACB) complex are a cautionary example for the world-wide spread of multi- and pan-drug resistant bacteria. Aiding the urgent demand for novel therapeutic targets, comparative genomics studies between pathogens and their apathogenic relatives shed light on the genetic basis of human-pathogen interaction. Yet, existing studies are limited in taxonomic scope, sensing of the phylogenetic signal, and resolution by largely analyzing genes independent of their organization in functional gene clusters. Here, we explored more than 3,000 *Acinetobacter* genomes in a phylogenomic framework integrating orthology-based phylogenetic profiling and microsynteny conservation analyses. We delineate gene clusters in the type strain *A. baumannii ATCC 19606* whose evolutionary conservation indicates a functional integration of the subsumed genes. These evolutionarily stable gene clusters (ESGCs) reveal metabolic pathways, transcriptional regulators residing next to their targets but also tie together subclusters with distinct functions to form higher-order functional modules. We shortlisted 150 ESGCs that either co-emerged with the pathogenic ACB clade or are preferentially found therein. They provide a high-resolution picture of genetic and functional changes that coincide with the manifestation of the pathogenic phenotype in the ACB clade. Key innovations are the remodeling of the regulatory-effector cascade connecting LuxR/LuxI quorum sensing via an intermediate messenger to biofilm formation, the extension of micronutrient scavenging systems, and the increase of metabolic flexibility by exploiting carbon sources that are provided by the human host. We could show experimentally that only members of the ACB clade use kynurenine as a sole carbon and energy source, a substance produced by

underlies figures and/or summary statistics is within the manuscript and its Supporting Information files. The large-scale dataset representing the sample of Proteobacteria obtained from NCBI RefSeq as well as the underlying raw Gower distance matrix are provided via figshare: https://doi.org/10.6084/m9.figshare.16910974.v1. Genomic data analyzed in this study are publicly available in the NCBI Reference Sequence Database (RefSeq) versions 87 and 204. For each selected genome, the assembly accessions are listed in S1 and S9 Tables, respectively. We obtained genome annotation files (*.gff and *_feature_table.txt), protein sequences and coding sequences (*.faa and *cds_from_genomic.fna), and genome sequence (*_genomic.fna) files from the database [ftp://ftp.ncbi.nlm.nih.gov/refseq/release/release-catalog/archive/]. The source code to our custom tool Vicinator is available under the GPL license 3.0 on Github (https://github.com/BIONF/Vicinator).

**Funding:** This study was supported by a grant by the German Research Foundation (DFG) in the scope of the Research Group FOR2251 "Adaptation and persistence of A. baumannii." Grant ID EB-285-2/2 to IE, AV 9/7-2 to BA, GO 2491/1-2 to SG, and WI 3272/3-2 to GW. MFF was supported by grants from the National Institute of Allergy and Infectious Diseases (grant R01AI144120). Cloud computational resources through the de.NBI cloud are granted by the German Bundesministerium für Bildung und Forschung grant FKZ 031A533B to AG. The funders had no role in study design, data collection and analysis, decision to publish, or preparation of the manuscript.

**Competing interests:** The authors have declared that no competing interests exist.

humans to fine-tune the antimicrobial innate immune response. In summary, this study provides a rich and unbiased set of novel testable hypotheses on how pathogenic *Acinetobacter* interact with and ultimately infect their human host. It is a comprehensive resource for future research into novel therapeutic strategies.

## Author summary

The spread of multi- and pan-drug resistant bacterial pathogens is a worldwide threat to human health. Understanding the genetics of host colonization and infection can substantially help in devising novel ways of treatment. Acinetobacter baumannii, a nosocomial pathogen ranked top by the World Health Organization in the list of bacteria for which novel therapeutic approaches are needed, is a prime example. Here, we have carved out the genetic make-up that distinguishes A. baumannii and its pathogenic next relatives from other and mostly apathogenic Acinetobacter species. We found a rich spectrum of pathways and regulatory modules that reveal how the pathogens have modified biofilm formation, iron scavenging, and their carbohydrate metabolism to adapt to their human host. Among these, the capability to metabolize kynurenine is particularly intriguing. Humans produce this substance to contain bacterial invaders and to fine-tune the innate immune response. But A. baumannii and closely related pathogens found a way to feed on kynurenine. This suggests that the pathogens might be able to dysregulate the human immune response. In summary, our study substantially deepens the understanding of how a highly critical pathogen interacts with its host, which substantially eases the identification of novel targets for innovative therapeutic strategies.

## Introduction

*Acinetobacter* is a physiologically and biochemically diverse genus of Gram-negative coccobacilli and most of its species are considered benign. But the genus also harbors the *Acinetobacter calcoaceticus-baumannii* (ACB) complex, a group of closely related human opportunistic pathogens [1,2] that account for the vast majority of severe hospital-acquired *Acinetobacter* spp. infections [3–7]. *Acinetobacter baumannii* is the most critical member of the ACB complex. On a global scale, this species alone signs responsible for up to 5% of the total bacterial infections in hospitals [8]. Many outbreaks worldwide can be attributed to one of eight genetically well distinguishable clonal complexes within the population of *A. baumannii*, all sharing the resistance against carbapenem [9,10]. By now, antibiotic resistance determinants against virtually all available antibiotics drugs are present in *A. baumannii* [11], and multi- or even pan-drug resistant strains are isolated from 44% of all patients with an *A. baumannii* induced infection [12]. At the same time, both the frequency and severity of infections have increased. Recent case studies report mortality rates of up to 70% [13–15] as well as growing numbers of epidemic outbreaks [16]. In recent years, significant advancements in the molecular characterization of drug resistance mechanisms have led to more informed drug administration schemes for hospitalized patients [6]. Still, the ease with which *A. baumannii* acquires resistance to novel antibiotics [12] makes it likely that resistant strains and their resistance determinants are going to spread at a faster pace than novel antimicrobials become available [17]. Moreover, community-acquired infections by members of the ACB complex begin to rise [6, 18]. As a consequence, *A. baumannii* ranks top in the WHO charts of pathogens for which drug development is most urgent [19].

A systemic understanding of how *A. baumannii* interacts with and infects their human host can lead to novel paths for antimicrobial treatments [20–22]. Three main approaches have been used to elucidate the molecular basis of *Acinetobacter* virulence. Candidate approaches have scanned for virulence factors previously characterized in other bacterial pathogens [4,7,23–27]. To extend the scope beyond pre-compiled virulence factor catalogs, a diverse set of genome-wide experimental approaches have been pursued. Among others, they assessed the effect of gene knockouts on the infection process (e.g. [14,28]), investigated transcriptional changes under conditions the bacterium encounters in the human host (e. g. [25,29,30]), studied adaptation evolution of bacteria inside the human host [31], and reconstructed the protein interaction network contributing to the understanding of bacterial antibiotic resistance mechanisms [32]. However, experiments are usually performed only on a small set of model strains (e.g. [33]), and the limited set of tested conditions cannot reflect the diversity of infection sites in the human body. Moreover, factors contributing only indirectly to virulence, such as metabolic pathways that facilitate the tapping of host resources [34], are hard to capture. Comparative genomics provide complementary evidences in the search of virulence related traits. The genus Acinetobacter encompasses to date 72 (validly) named and mostly nonpathogenic species [35] isolated from habitats that range from floral nectar to animals [36]. This diversity represents a perfect setup to identify genomic changes that correlate with the evolutionary emergence of the pathogenic potential [37]. Thus far, comparative genomics studies have begun to shed light on the general evolution of the genus [11] and the clonal epidemiology [38] of *A. baumannii*. They indicated that a major driver of *A. baumannii*'s success as a pathogen is its remarkably flexible genome [11,39] which is characterized by high mutation rates [40] paired with the ability to acquire new, or alter the structure or expression of existing genes [13–15]. This promotes a rapid adaptation to novel and adverse environmental conditions, as well as the spread of antimicrobial resistance determinants [41,42]. However, against intuition, most of the known virulence determinants were found also in the nonpathogenic members of the genus (e.g. [4,43–45]). Thus, it is still largely unknown which genetic changes correlate with the emergence of the ACB complex as opportunistic human pathogens, and the genetic basis underlying the adaptation of *A. baumannii* to the human host largely remains to be understood [46].

Here, we exploit the availability of thousands of *Acinetobacter* spp. genomes in the public databases (NCBI RefSeq; [47]) to shed light on the evolution of the pathogenic ACB complex at a resolution that extends from a genus-wide overview to the level of individual clonal lineages within *A. baumannii*. For the first time, we integrate genus-wide ortholog searches with analyses of gene order conservation providing a highly-resolved view on the joint evolutionary fate of neighboring genes using the type strain ATCC 19606 as a reference. This revealed 150 evolutionarily stable gene clusters (ESGC$_{ACB}$) that are prevalent in the ACB complex and rare or absent in the other members of the genus. The functional annotations of these ESGCs provide insights into the genetic and functional specifics of a clade comprising mostly pathogens, and thus direct the focus to key processes likely relevant for the adaptation of the bacterium to the human host. We find that the ACB complex acquired novel genetic modules for the regulation and formation of biofilms, for the scavenging of micronutrients, and have substantially extended their capabilities to exploit a diverse set of carbon sources.

## Results

The ability to infect humans emerged in the course of *Acinetobacter* spp. evolution and is a hallmark of the ACB complex. Here, we exploited the full diversity of *Acinetobacter* genomes available in the public databases that were available at the onset of the study (S1 Table). We use

this resource to trace changes in the *Acinetobacter* pan-genome that correlate with the manifestation of pathogenicity in the ACB complex. To make the analyses computationally tractable we devised a two-stage strategy.

In the priming stage, we determined the evolutionary relationships within the *Acinetobacter* pan-genome with the orthology inference tool OMA [48]. Because the computational complexity of the OMA ortholog search scales exponentially with the numbers of genes in the pan-genome, we compiled a representative set of strains (Set-R) for this analysis. In brief, we considered all available type, reference and representative genomes, as well as all validly named species for which a genome sequence was available at the study onset. This set was filled to a total number of 232 strains by adding further genomes to maximize the phylogenetic diversity of the taxon set (see Methods and Fig A in S1 Text). The corresponding strains together with genome assembly statistics, and, where available, the origins of the isolates are summarized in S2 Table. Members of the ACB complex harbor, on average, 14% more genes than other members of this genus (students t-test—p<0.001; S1A Fig). Gene counts were highly correlated with genome lengths (spearman, $\rho = 0.98$), and neither the difference in genome length nor the number of encoded genes was significantly correlated with the assembly status (Completeness status "Complete" vs. Others, Kruskal-Wallis, p = 0.311) (see S1B Fig for further information). The Set-R pan-genome comprises 22,350 orthologous groups harboring 783,306 proteins; 16,000 proteins remained singletons (see S1C Fig for a graphical representation). Rarefaction analyses revealed that the pan-genomes of the entire genus, the ACB complex, and *A. baumannii* are open (Fig B in S1 Text). 889 genes represent the core genome of *Acinetobacter* (S3 Table and S1 Text: Section Core-genome reconstruction). Eventually, we tentatively annotated gene function in the Set-R pan-genome by linking the individual genes to COGs [49], to KEGG KOs [50], to entries in the virulence factor databases PATRIC [23] and VFDB [51], and by predicting their subcellular localization.

In the extension stage, we used a targeted ortholog search to complement the orthologous groups from the SET-R analysis with sequences from the remaining 2,820 *Acinetobacter* genomes (Set-F).

## The Acinetobacter-Dashboard

Acinetobacter research worldwide benefits from the FAIR principle where scientific data is findable, accessible, interoperable and reproducible [52]. As a first step in this direction, we have developed the web application Aci-Dash (https://aci-dash.ingress.rancher.computational.bio/; Fig 1). For each strain, the user can obtain information about the sample origin and get access to all genes annotated in the respective genome together with an overview of its abundance in the other 231 strains. For example, this allows the rapid identification of genes that are specific to a strain, a clade, or that are part of the core genome. Moreover, genes can be sub-selected based on their genomic position, which allows to explore the phylogenetic profiles of neighboring genes. Interactive plots make it straightforward to retrieve further information about individual or groups of genes, such as their assignment to COG or KO groups, or their representation in virulence databases (see above). Thus, Aci-Dash is the first web-based platform to interactively browse and explore the Acinetobacter pan-genome that is spanned by the 232 strains of Set-R.

## Consistency-based phylogeny of the genus *Acinetobacter*

To establish a stable phylogenetic backbone for our analysis, we reconstructed the maximum likelihood evolutionary relationships of the taxa in Set-R and Set-F, respectively, from three non-overlapping partitions of the 889 core genes. The majority-rule consensus phylogenies

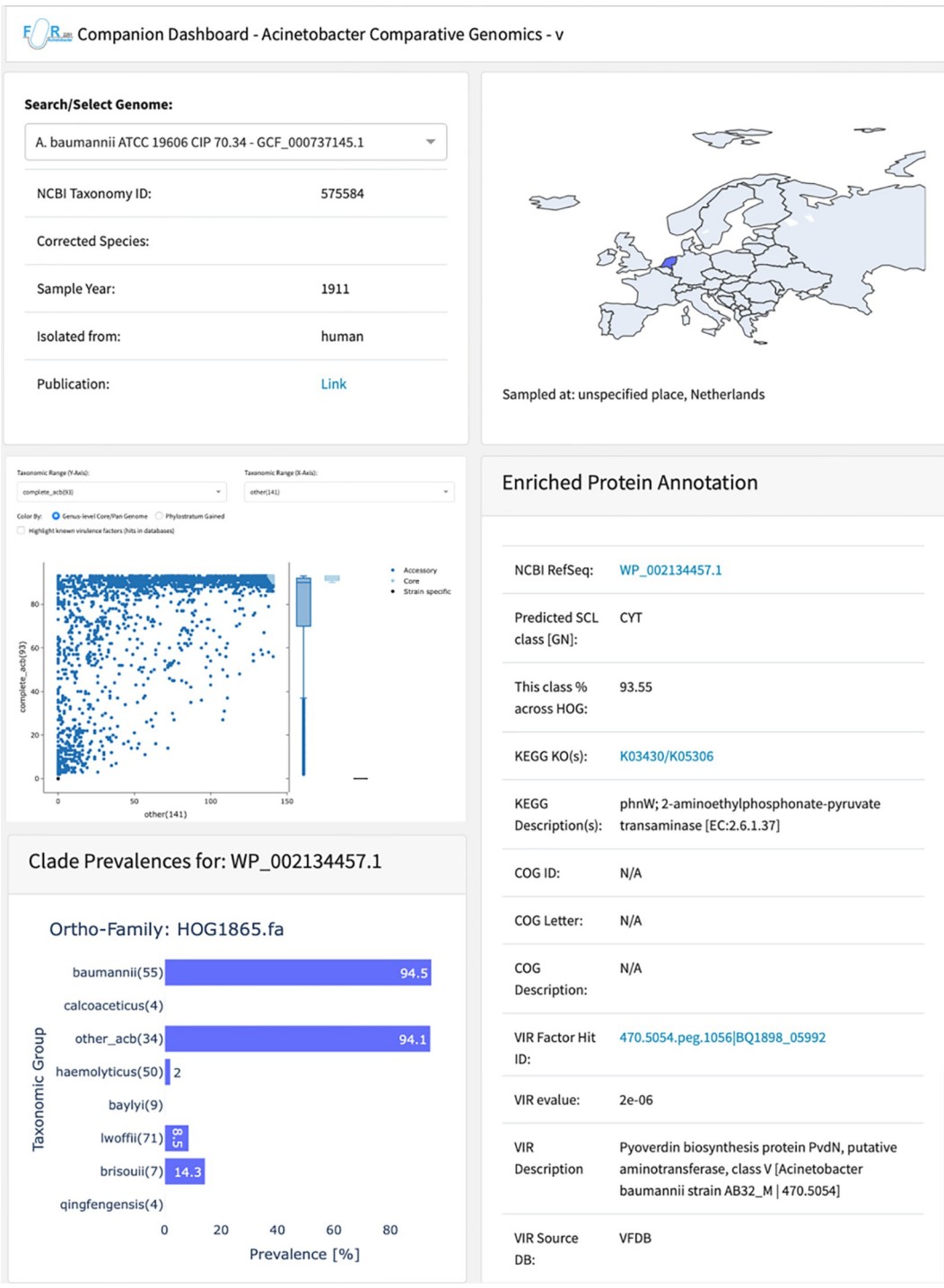

**Fig 1. Aci-Dash–Interactive exploration of phylogenetic abundance patterns and accessory annotations for the Set-R pan-genome.** For each of the 232 strains, Aci-Dash provides further details about year and site of sampling. The map was generated with Plotly for Python with a basemap from Natural Earth (https://www.naturalearthdata.com). The interactive scatter plot reveals, for each gene of the selected strain, the abundance of orthologs in a user-defined in- and outgroup. For each gene individually, the ortholog abundance can be resolved on a clade level (cf. Fig 2), and further information including functional annotation transfer from KEGG and COG as well as known virulence factors is displayed.

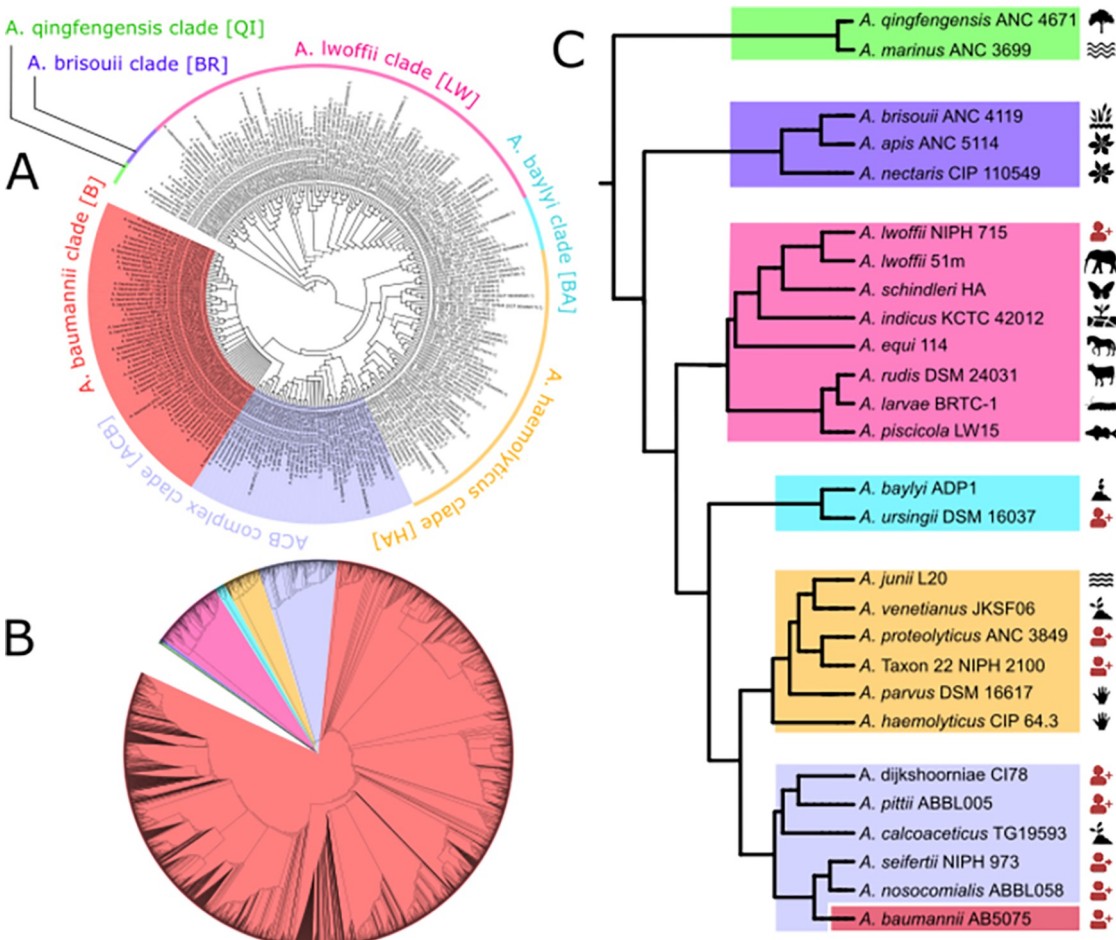

**Fig 2. The phylogeny of the genus Acinetobacter.** (A) Majority-rule consensus phylogeny of 232 Acinetobacter strains represented in SET-R. Solid branches are supported by all, and hatched branches by two out of three trees. A high-resolution image of this tree is provided in S2 Fig. (B) The maximum likelihood tree for all 3052 taxa in Set-F. Colored clades represent the same clades as in A). A high-resolution image of this tree is provided in S3 Fig. (C) The evolutionary backbone of the Acinetobacter genus with exemplary strains as clade representatives. The color scheme resembles that of Fig 2A. The pictograms next to the leaf labels indicate the sampling source of the particular strain. Red pictograms signal a strain that was isolated from an infected patient. The clipart used in this figure has been dedicated to the public domain (CCO 1.0 Universal) or was self-generated.

from the three trees each (Set-R–Fig 2A; Set-F–Fig 2B; for higher resolution versions see S2 and S3 Figs, respectively) reveal that all named species (at the time of download) as well as the members of the ACB complex are consistently placed into monophyletic clades. Incongruencies between the three partition trees are confined to the branching order within individual species, and here mainly within the densely sampled *A. baumannii* and *A. pittii*. This indicates that genetic recombination, which is most likely the source of the incongruent phylogenetic signal [53], is common enough only within species to interfere with phylogenomic reconstructions based on hundreds of genes [54]. Across the genus, we detected and corrected individual taxonomic assignments that are at odds with the phylogenetic placement of the taxa, and most likely indicate mislabeled strains ([55,56]). Specifically, we corrected 16 of such instances within the ACB clade, of which ten were wrongly classified as *A. baumannii* according to NCBI RefSeq. In turn, 60 out of 182 genomes with an unknown taxonomic assignment were placed within the ACB clade (see S4 Table for species and clade assignments including the corrections). Interestingly, a comparison of the average pair-wise nucleotide identity (ANI) across

all genomes within Set-R revealed that at least two genomes placed into the ACB clade cannot be associated with any known species (ANI <95%, *cf*. S4 Fig). This indicates that the full species diversity of the ACB complex is not yet fully charted.

To ease the integration of the phylogenetic information into the following sections, we used one species each to name the individual clades in the *Acinetobacter* phylogeny (Fig 2A).

### Lifestyle and host switches during *Acinetobacter* evolution

The two earliest branching clades, named after *A. qingfengensis* (QI) and *A. brisouii* (BR), respectively, solely comprise environmental species. *A. apis*, which was isolated from bees [57], appears as an exception. While *Acinetobacter* species are sporadically observed in the bee gut, they are not considered part of the gut microbiome [58]. Instead, they likely represent environmental bacteria that were taken up by the bee with the food [59]. Thus, the capability to colonize animals evolved later and most likely in the ancestral species prior to the split of the *A. lwoffii* (LW) clade (Fig 2B). Usually, members of the LW clade are non-pathogenic (Fig 2C). Repeated cases of human infection have only been reported for individual strains that mainly group with *A. lwoffii* and *A. radioresistens*, which manifested in vascular catheter-induced bloodstream infections with a low mortality rate [60]. Thus, human infection is likely an exception rather than the rule for this clade. In the species that diverged after *A. baumannii* last shared a common ancestor with *A. baylyi* (AB clade) and with *A. haemolyticus* (HA clade), we find increasingly often human pathogens. This suggests a progressive adaptation to humans as a host [61]. The monophyletic ACB complex (ACB clade) subsumes the *A. pittii clade* (PI), the *A. nosocomialis* clade (NO), and the *A. baumannii* clade (B). Its members are all potentially pathogenic, although *A. calcoaceticus* has been very rarely been seen in the context of human infection. It can be speculated that the few reported cases were due to a miss-classified strain from a different species (see subsection "Consistency-based phylogeny of the genus Acinetobacter" above). Thus, *A. calcoaceticus* substantially reduced its pathogenic potential if not lost it completely [27].

### Functional innovation in the Set-R pan-genome

We next scanned the Set-R pan-genome for gains of function on the lineage towards contemporary *A. baumannii*. We tentatively stratified the pan-genome by assigning each orthologous group to the inner node in the *Acinetobacter* phylogeny that represents the last common ancestor (LCA) of the two most distantly related taxa in that group (Fig 3A). For each node, we then determined the set of significantly enriched gene functions using the gene ontology (GO) terms from the sub-ontology Biological Process. Because genes with a sparse phylogenetic distribution cannot drive the shared phenotype of a clade, we confined the GO term enrichment analysis to only the subset of orthologous groups, where orthologs were detected in at least half of the subsumed taxa of a node (Fig 3B and Tables 1 and S5). This revealed several processes with potential relevance for pathogenicity, e.g. cell adhesion, siderophore biosynthesis, and response to oxidative stress. However, reproducing previous observations [4, 43–45], the corresponding genes were assigned to nodes in the phylogeny that predate the emergence of the ACB clade. Only three significantly enriched GO terms were assigned to the node representing the LCA of the ACB clade (Table 1), and no term was significantly enriched in the genes private to *A. baumannii*.

### Evolutionary and functional units in the ATCC 19606 gene set

The GO term enrichment analysis revealed only a weak signal for the gain of biological processes that can be directly connected to bacterial virulence in the evolutionarily younger nodes

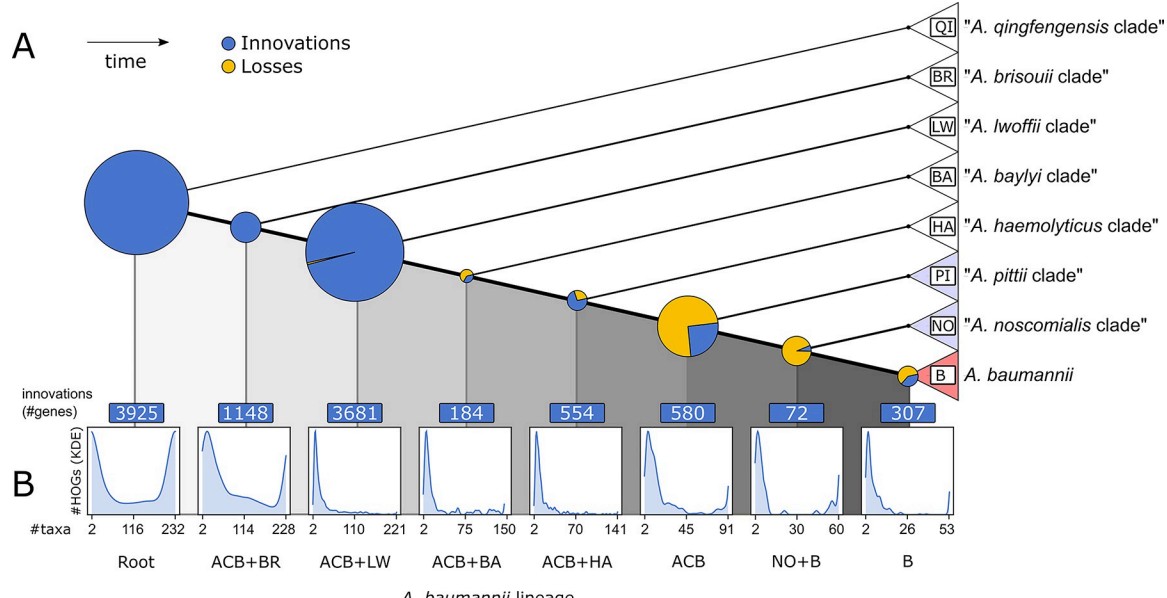

**Fig 3. A phylogenetic stratification of the Acinetobacter pan-genome.** (A) Pie charts on the internal nodes indicate numbers of genes added (blue) or lost (yellow) from the respective pan-genome where the diameter is proportional to the sum of both gained and lost genes. Clades resemble those from Fig 2. (B) For each node, histograms represent the number of descending taxa an added gene is represented in. The minimum value is 2, the maximum value is the number of taxa subsumed in the corresponding clade.

of the *Acinetobacter* phylogeny. However, two factors confound the analysis. Only about two-thirds of all unique sequences in the *Acinetobacter* spp. pan-genome are annotated with a GO term and many of the genes with only unspecific terms. Moreover, the sporadic presence of orthologs to individual genes in taxa outside the ACB clade, e.g., as a result of recombination, confounds the phylogenetic stratification of the pan-genome. To increase both power and resolution, we widened the focus and traced the emergence and evolution of gene clusters in *Acinetobacter* spp. using the type strain *A. baumannii* ATCC 19606 as a reference. For each gene in the type strain, we integrated its node assignment, the abundance of orthologs within *A. baumannii*, within other members of the ACB clade, and within the remaining taxa in SET-R, respectively. Where applicable, we added information about the overrepresented GO terms (Fig 4). As a recurrent theme, we observed that genes linked to the same overrepresented GO term reside adjacent to genes with highly similar phylogenetic abundance patterns, despite being occasionally assigned to different nodes in the phylogeny. We subsequently connected neighboring genes with significantly similar abundance patterns across the entire genome of *A. baumannii* ATCC 19606 to form candidate clusters (S6 Table). A candidate cluster was then propagated to represent an evolutionarily stable gene cluster (ESGC), which likely forms a functional entity, if its gene order was found conserved within the ACB clade. We then short-listed 150 ESGC$_{ACB}$ that are abundant among members of the ACB clade but rare or even absent in other taxa (S6 Table and S5 Fig; see Figs C and D in S1 Text). As the last step, we manually validated the automatic ESGC$_{ACB}$ assignment for 44 clusters whose sets of genes were each functionally annotated to a level that allowed us to infer general cluster function (Fig 5), and for further 10 clusters with an unknown function but that are almost exclusively found in the ACB clade (Fig 6). These 54 ESGC$_{ACB}$ can be broadly distinguished into four categories: The cluster comprises (i) a known operon or metabolic gene cluster (e.g., ESGC$_{ACB}$-0622, phenylacetate metabolism), (ii) a group of functionally related genes that are linked to genes, whose functional link, e.g. acting as a transcriptional regulator, was unknown thus far

**Table 1. Selection of overrepresented GO terms in the assigned gene sets of the inner nodes (The full list of enriched terms is provided in S5 Table, Table 1).**

| Node | GO | Biological Process | p_cor[1] | RT[2] | RB[3] | d[4] | SC[5] | SA[6] | %B[7] | %ACB[8] | %nACB[9] |
|------|----|--------------------|----------|-------|-------|------|-------|-------|-------|---------|----------|
| **ACB+BR (n = 230)** | GO:0018189 | pyrroloquinoline quinone biosynthesis | $<10^{-7}$ | 8.6 | 5.0 | 9 | 192 | 5 | 97.8 | 98.7 | 17.0 |
| | GO:0006855 | drug tm transport | $<10^{-7}$ | 2.4 | 9.5 | 5 | 53 | 1 | 96.4 | 96.8 | 23.4 |
| | GO:0015810/3 | aspartate/L-glutamate tm transport | $<10^{-7}$ | 4.3 | 2.5 | 10 | 96 | 2 | 96.4 | 97.8 | 24.8 |
| | GO:0071705 | nitrogen compound transport | $<10^{-7}$ | 2.8 | 9.1 | 4 | 62 | 1 | 100 | 100 | 27.7 |
| | GO:0019439 | aromatic compound catabolism | $5.2*10^{-3}$ | 2.2 | 14.0 | 4 | 49 | 2 | 92.7 | 95.7 | 28.4 |
| | GO:0006631 | fatty acid metabolism | $<10^{-7}$ | 3.3 | 6.5 | 7 | 73 | 1 | 89.1 | 91.4 | 36.2 |
| | GO:0019619 | 3,4-dihydroxybenzoate catabolism | $<10^{-7}$ | 3.2 | 10.6 | 8 | 72 | 1 | 90.9 | 92.5 | 39.0 |
| | GO:0007155 | cell adhesion | $<10^{-7}$ | 4.6 | 10.6 | 2 | 103 | 1 | 98.2 | 98.9 | 40.4 |
| | GO:0010124 | phenylacetate catabolism | $<10^{-7}$ | 10.4 | 8.5 | 8 | 233 | 4 | 89.6 | 90.6 | 45.7 |
| **ACB+LW (n = 223)** | GO:0031388 | organic acid phosphorylation | $<10^{-7}$ | 2.9 | 2.4 | 6 | 54 | 1 | 98.2 | 98.9 | 14.9 |
| | GO:0009437 | carnitine metabolism | $3.1*10^{-3}$ | 0.2 | 0.1 | 6 | 3 | 1 | 94.5 | 88.2 | 22.0 |
| | GO:0071705 | nitrogen compound transport | $<10^{-7}$ | 4.4 | 10.5 | 4 | 83 | 2 | 99.1 | 94.6 | 23.4 |
| | GO:0036104 | Kdo2-lipid A biosynthesis | $<10^{-7}$ | 2.4 | 3.8 | 8 | 45 | 1 | 94.5 | 91.4 | 29.8 |
| | GO:0009435 | NAD biosynthesis | $<10^{-7}$ | 3.4 | 15.7 | 11 | 65 | 1 | 89.1 | 91.4 | 30.5 |
| | GO:0009116 | nucleoside metabolism | $<10^{-7}$ | 3.5 | 13.0 | 6 | 67 | 1 | 96.4 | 95.7 | 31.2 |
| | GO:0055085 | tm transport | $<10^{-7}$ | 24.2 | 149.4 | 4 | 459 | 7 | 90.2 | 89.1 | 38.7 |
| | GO:0019557/6 | histidine catabolism to glutamate and formate/formamide | $<10^{-7}$ | 9.9 | 6.1 | 10 | 188 | 3 | 95.8 | 97.5 | 41.6 |
| | GO:0006351 | transcription, DNA-templated | $<10^{-7}$ | 86.9 | 320.1 | 9 | 1645 | 21 | 95.5 | 93.7 | 41.8 |
| | GO:0019290 | siderophore biosynthesis | $<10^{-7}$ | 8.0 | 4.8 | 8 | 170 | 3 | 95.8 | 72.8 | 41.8 |
| **ACB+BA (n = 152)** | GO:0045150 | acetoin catabolism | $<10^{-7}$ | 33.6 | 0.8 | 7 | 36 | 1 | 100 | 100 | 14.9 |
| | GO:0019290 | siderophore biosynthesis | $<10^{-7}$ | 29.9 | 5.6 | 8 | 32 | 1 | 96.4 | 97.8 | 22.0 |
| | GO:0006351 | transcription, DNA-templated | $<10^{-7}$ | 76.5 | 0.03 | 9 | 82 | 2 | 84.5 | 84.9 | 26.2 |
| **ACB+HA (n = 143)** | GO:0022904 | respiratory electron transport chain | $<10^{-7}$ | 45.9 | 5.39 | 5 | 44 | 0 | 90.9 | 79.6 | 11.3 |
| | GO:0006351 | transcription, DNA-templated | $<10^{-7}$ | 78.3 | 314.8 | 9 | 75 | 0 | 90.9 | 91.4 | 27.0 |
| **ACB (n = 93)** | GO:0009372 | quorum sensing | $<10^{-7}$ | 8.8 | 0.3 | 3 | 13 | 2 | 86.4 | 90.3 | 0.0 |
| | GO:0006351 | transcription, DNA-templated | $<10^{-7}$ | 98.9 | 318.1 | 9 | 146 | 5 | 92.0 | 87.1 | 0.0 |
| | GO:0006979 | response to oxidative stress | $4.7*10^{-5}$ | 9.5 | 17.4 | 3 | 14 | 1 | 89.1 | 78.5 | 0.0 |

[1] corrected p-value

[2] Ratio of genes with term in test set of that node ($\times 10^{-3}$)

[3] Ratio of genes in background set of that node ($\times 10^{-4}$)

[4] Depth of term in GO tree

[5] Count of subjects with term in study set of that node

[6] Number of ortholog groups associated with proteins in the study set. For this table, redundancy was reduced by selecting the lowest depth GO terms if any two GO terms had non-empty intersections between their sets of protein sequence identifiers or associated ortholog group identifiers.

[7] Mean prevalence of associated ortholog groups in *A. baumannii*

[8] Mean prevalence of associated ortholog groups in the ACB clade

[9] Mean prevalence of associated ortholog groups in *Acinetobacter* spp.

(e.g., ESGC$_{ACB}$-0162), (iii) super-clusters connecting two or more clusters with distinct functions to form a higher-order functional unit (ESGC$_{ACB}$-0394, *adeFGH* and *hisQMP*), and (iv) clusters of genes with an unknown function (e.g., ESGC$_{ACB}$-0503 to 0510; Fig 6). We hypothesize that the adaptation of pathogenic *Acinetobacter* species to their human host is mirrored in the various functions that these ESGC$_{ACB}$ convey. In the following sections, we highlight a selection of clusters related to persistence, micronutrient acquisition, and the evolution towards nutritional flexibility (see S1 Text for ESGCs that are not discussed in the main manuscript).

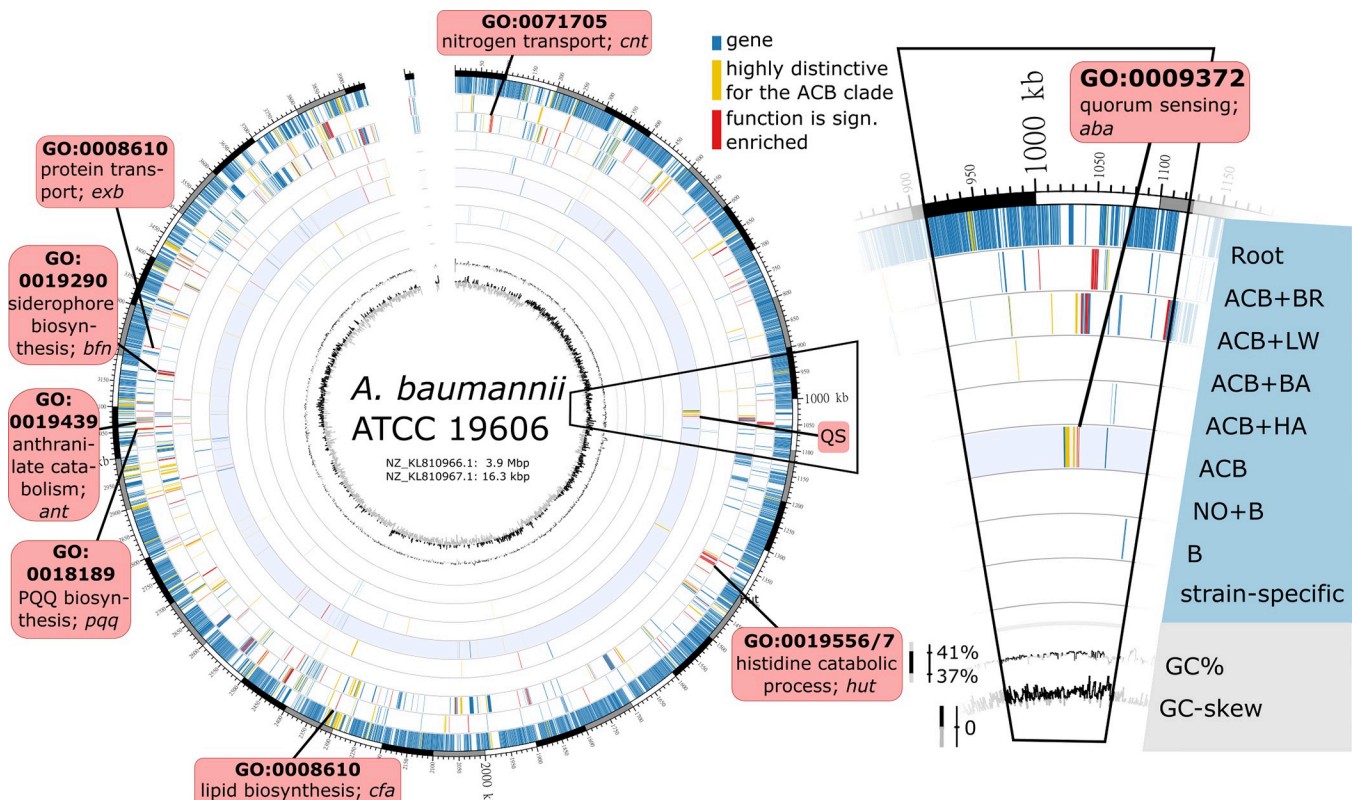

**Fig 4. Phylogenetic strata in the protein-coding gene set of Ab. ATCC 19606.** Each protein-coding gene was assigned to one of the nine layers specified in the inlay to the right (cf. Fig 3). All genes annotated with GO terms that were significantly enriched in the individual layers are colored in red. Genes assigned to orthologous groups which were ranked within the 10 percent groups with highest retention differences (RD > = 0.708, n = 355) across the ATCC 19606 gene set are colored in yellow. Red boxes highlight selected gene loci where neighboring genes contributed to the enrichment of the same GO term (biological processes).

## The evolutionary emergence of the ACB clade coincides with changes in quorum sensing and biofilm formation

Quorum sensing and biofilm formation are key determinants of *A. baumannii* virulence [62–64]. Both functions are represented by ESGC$_{ACB}$-0162 (14 genes) and 0410 (8 genes). ESG-C$_{ACB}$-0162 represents the regulatory module of this process. It harbors the Lux-type quorum sensing system (QS$_{Lux}$), which regulates motility and biofilm formation in *A. baumannii*, and additionally, a biosynthetic gene cluster containing a non-ribosomal peptide synthetase here referred to as NRPS cluster. Both genes of QS$_{Lux}$, *abaI* and *abaR*, are separated by a short gene that was tentatively named *abaM* (Fig 7A; see also Figs E and F in S1 Text). This three-gene architecture is conserved in the ACB clade, and it is common in *Burkholderia* spp. [65], where the intervening short gene acts as a negative regulator of the QS$_{Lux}$ system [66]. Recently, initial evidence emerged that AbaM in the *A. baumannii* strain AB5075 indeed regulates quorum sensing and biofilm formation [67]. In the light of our results, we propose that AbaM is an understudied modulator of quorum sensing in the entire ACB clade.

The NRPS cluster (Fig 7A) produces a three-amino acid lipopeptide, Ac-505 [64], which likely plays a central role in regulating bacterial motility and biofilm formation [68]. Disrupting its biogenesis alters the expression of numerous factors involved in biofilm formation and surface adherence [69], in particular the chaperon-usher pili (CUP) and the archaic chaperon-usher pili (CSU). Consequently, host cell adhesion and virulence of *A. baumannii* are

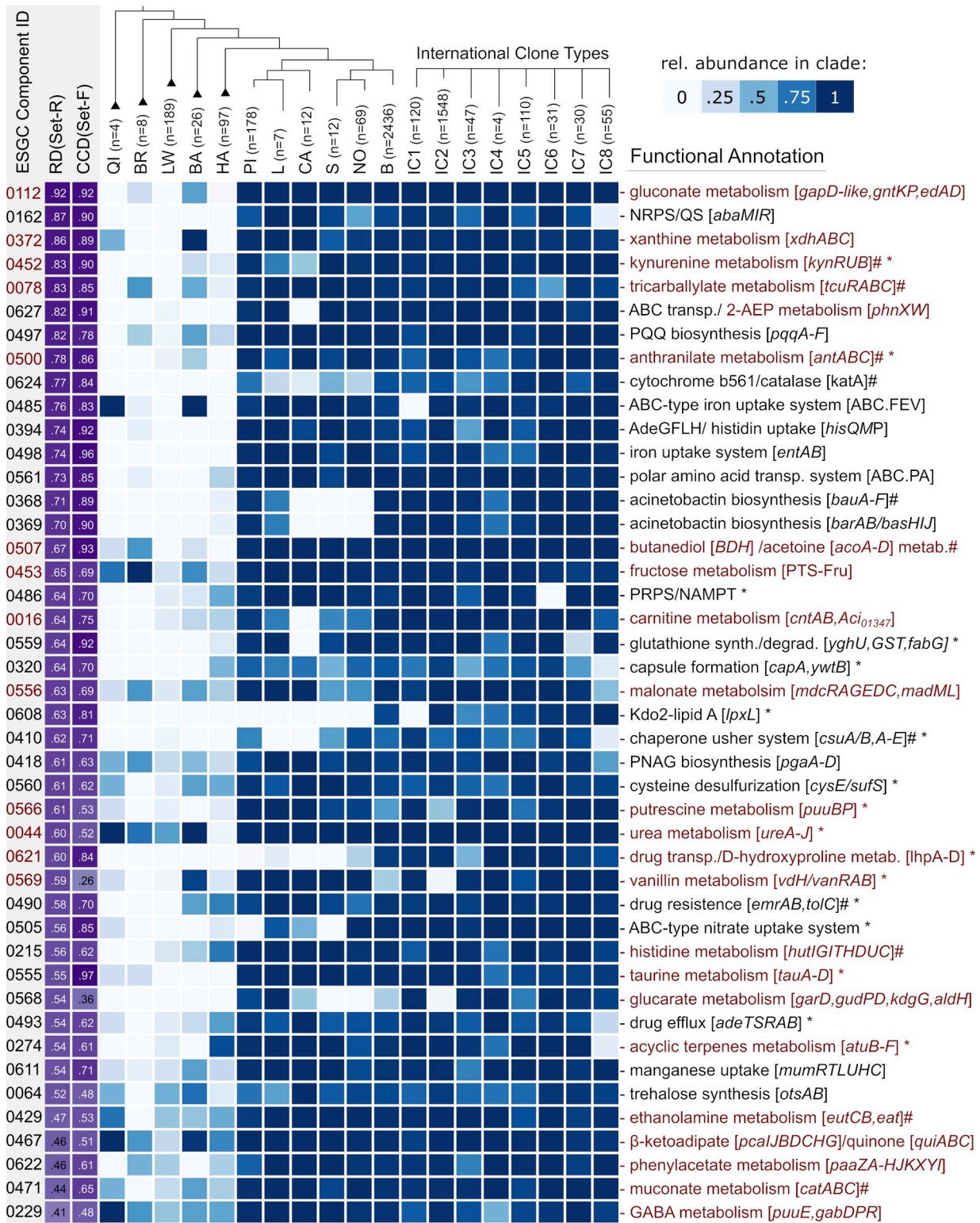

**Fig 5. Phylogenetic profiles of 44 functionally annotated ESGC_ACB across the Set-F.** Shown are the selection of ESGC_ACB with the highest abundance difference between the ACB clade and non-ACB taxa (RD). Ids refer to the corresponding connected components provided in S6 Table. Gene clusters are ranked by decreasing RD. The CCD (cluster conservation difference, violet) is calculated as the difference between the relative fractions of ACB-genomes minus non-ACB genomes where the cluster is present. The heat map informs about the fraction of taxa per clade harboring the ESGC_ACB ranging from 0 (white) to 1 (dark blue). The total numbers of subsumed taxa per clade are given next to the leaf

label in the above tree. IC1-8 represent the cluster abundance in the 8 international clones of A. baumannii. Clusters associated with metabolic pathways are highlighted in red. "#" marks an ESGC where we curated cluster boundaries based on literature evidence and confirmed cluster conservation using microsynteny (cf. Fig 4B). '*' marks ESGC_ACB that encompass additional genes not covered by the functional annotation. Abbreviations: NRPS = non-ribosomal peptide synthesis; QS = quorum sensing; 2-AEP = 2-aminoethylphosphonate; PNAG = polymeric β-1,6-linked N-acetylglucosamine; NAMPT = nicotinamide phosphoribosyl transferase; PRPS = 5-phosphoribosyl pyrophosphate synthetase; GABA = gamma-aminobutyric acid; PQQ = pyrroloquinoline quinone.

substantially reduced. Here, we provide first-time evidence that the evolutionary fate of the NRPS cluster is intimately intertwined with that of the QS_Lux cluster. We found that the rare strain-specific loss of the QS_Lux-cluster determines the loss of the NRPS cluster, which implies that they not only form an evolutionary but also a functional unit. Interestingly, strains lacking ESGC_ACB-0162 are not randomly distributed. Most prominently, the cluster is missing in almost all (48/55) *A. baumannii* strains representing the international clone (IC) 8 (Fig 5). The formation of a higher-order module comprising the QS genes and an NRPS biosynthetic gene cluster is a repeated scheme during bacterial evolution. For example, the methane-oxidizing bacterium *Methylobacter tundripaludum* harbors an NRPS biosynthetic gene cluster that was integrated between the *abaI* and *abaR* orthologs. And the production of the corresponding extracellular factor is under control of the QS cluster [70]. NRPS-dependent molecules have been implicated to mediate interspecific communications across kingdoms both in symbiotic and pathogenic communities. In *P. aeruginosa*, the interplay of N-acyl-L-homoserine lactone-dependent quorum-sensing signaling and an NRPS-dependent biosynthesis of bacterial cyclo-dipeptides (CDPs), which act as auxin signal mimics, modulates the communication to its host plant *Arabidopsis thaliana* [71]. It can be speculated that ESGC_ACB-0162 may similarly coordinate the communication between the bacteria and their human host.

ESGC_ACB-0410 harbors the Csu cluster responsible for biofilm formation on abiotic surfaces via archaic chaperon-usher pili [72,73] (Fig 7B) together with a transcriptional regulator

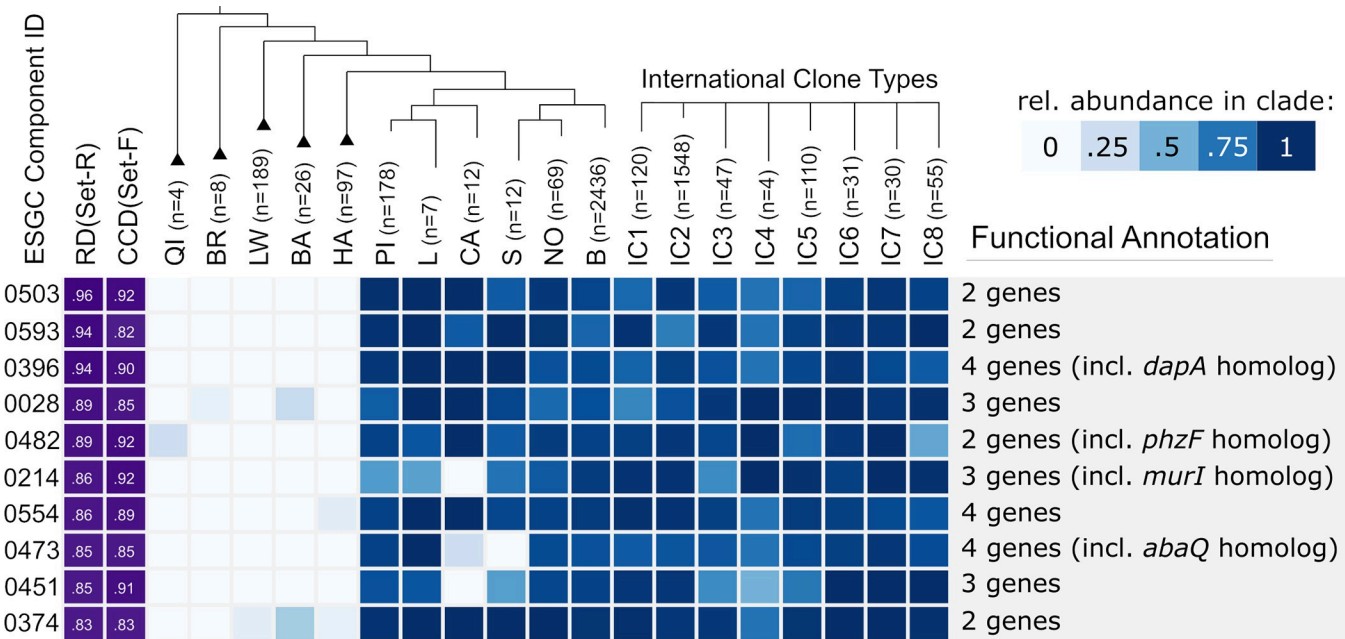

**Fig 6. Top 10 ESGC_ACB with unknown cluster function.** The genes of these clusters are mostly annotated as 'hypothetical' or "DUF-containing protein", i.e. proteins with annotated with a domain of unknown function. Due to their stable microsynteny and high prevalence across the ACB clade, they are highly interesting candidates for further functional characterization. The figure layout follows Fig 5.

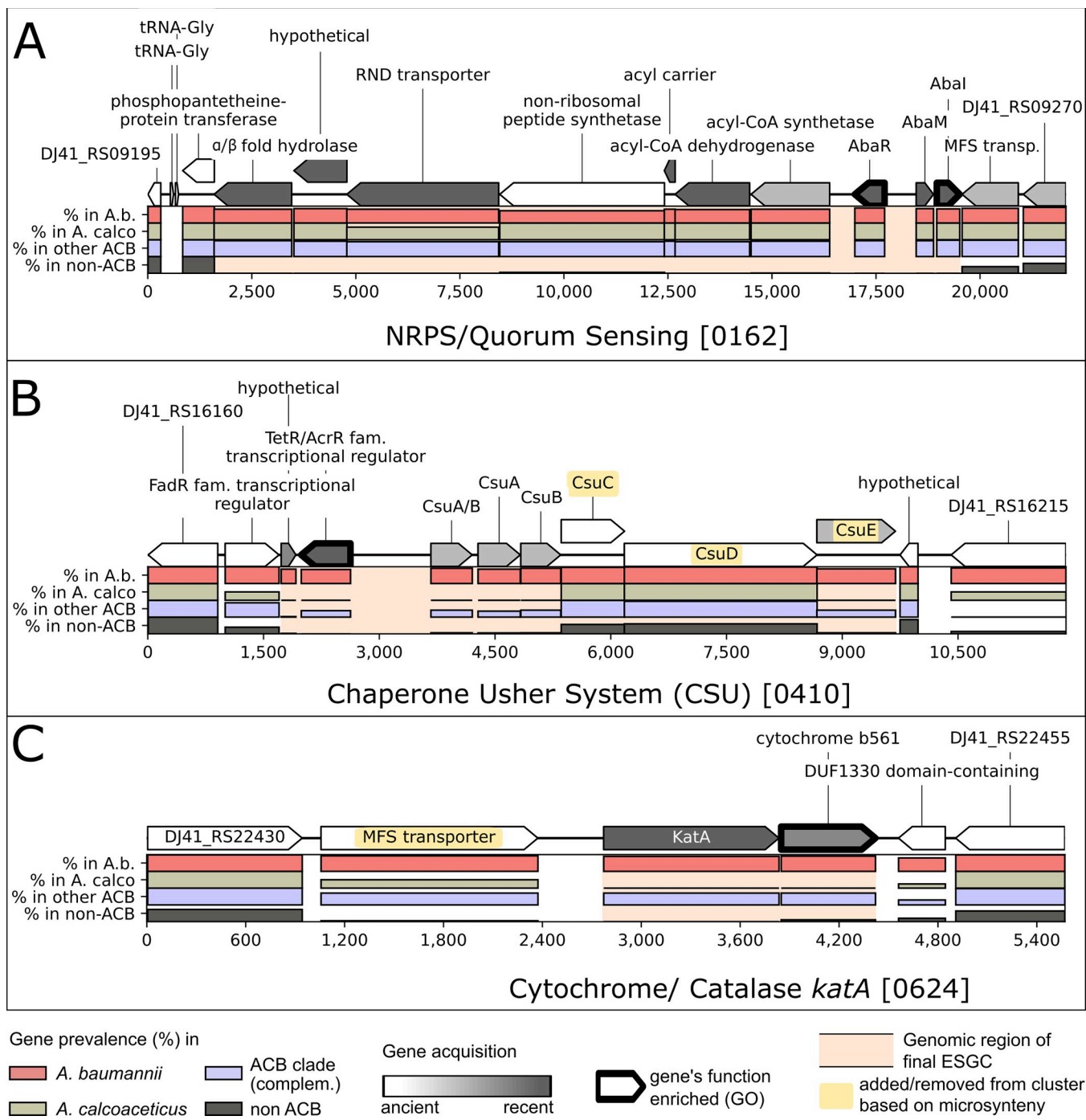

**Fig 7. Three examples for ESGC$_{ACB}$ in the genome of Ab ATCC 19606.** Bar plots indicate relative abundance within the selected taxonomic groups (see legend). The ESGC$_{ACB}$ are embedded into two flanking protein-coding genes on each side that are not part of the cluster. Cluster boundaries are indicated by a pink background. Genes indicated in yellow have been either added to or excluded from the automatically generated clusters based on microsynteny analyses across Set-R. (A) ESGC$_{ACB}$− 0162 unites AbaR and AbaI (quorum sensing) with a non-ribosomal peptide synthetase cluster upstream. (B) ESGC$_{ACB}$− 0410 encompasses the cluster necessary for the Csu pilus formation. Note the deviating abundance pattern for CsuC and CsuD, which is due to the presence of prpC and prpD, two paralogous genes from the photoregulated pilus ABCD (prpABCD), in the respective orthologous groups. Microsynteny analyses confirmed that the entire Csu cluster forms one evolutionary unit (S1 Data id:0410). (C) ESGC$_{ACB}$− 0624 harbors a catalase and a cytochrome b561. Although the MFS transporter shares a similar abundance pattern across the taxa of Set-R, this transporter is not evolutionarily stably linked to the other two genes (S1 Data id:0624).

of the TetR/AcrR family (TFTRs) (S5 Fig id:0410). TFTRs represent one-component systems that regulate a broad variety of cellular processes in bacteria, among them many that are related to virulence such as efflux pump expression and biofilm formation [74,75]. Notably, they are often encoded alongside their target operons. To the best of our knowledge, regulation of the Csu cluster via an adjacent TFTR has never been reported. Thus, next to the two-component systems BfmRS [76] and GacSA [77], a third hitherto undescribed one-component system, seems to be involved in regulating the formation of Csu pili.

With few exceptions, ESGC$_{ACB}$-0162 and ESGC$_{ACB}$-0410 share similar abundance patterns (cf. Fig 5). This is in line with the finding that the regulation of the Csu cluster is under the direct control of Ac-505 [28,69]. Thus, Ac-505 likely acts as a modulator between biofilm formation on abiotic and biotic surfaces. However, contrary to the QS$_{Lux}$—NRPS supercluster, the Csu cluster was lost multiple times independently in the ACB clade, e.g. in the CA and the L clades (Fig 5). Given its terminal position in the regulator-effector cascade, this indicates a lineage-specific fine-tuning of biofilm formation. Interestingly, within *A. baumannii*, we find that all 55 IC8 strains in our dataset lack both the QS$_{Lux}$—NRPS supercluster and the Csu cluster, which indicates substantial changes in the way how IC8 strains regulate biofilm formation.

## KatA–An ACB clade specific catalase

The genome of *A. baumannii* ATCC 19606 harbors five putative catalases: *katA*, *katE*, *katE-like*, *katG*, and *katX* [25,78]. Note, that both Sun et al. [78] and Juttukonda et al. [25,78] refer to a catalase labeled *katE*. Despite the same names, the studies refer to different genes (locus tags A1S_1386/A1S_3382 and DJ41_RS22765/DJ41_RS10660 in *A. baumannii* ATCC 17978 and ATCC 19606, respectively). We, therefore, re-named katE of Juttukonda et al. to *katE*-like. *KatA* is the only catalase that is exclusively found in the ACB clade. The corresponding gene resides in a cluster next to a putative MFS transporter and a cytochrome b561 (ESGC$_{ACB}$-0624, Fig 7C). The KatA cluster is highly conserved in all species of the ACB clade except *A. calcoaceticus*, where it has been lost (cf. Figs 5 and 7C). Upon host infection, both neutrophils and macrophages recruit radical oxygen species (ROS) for bacterial clearance [79, 80], and thus ROS defense mechanisms are an essential contributor to bacterial virulence. However, an initial investigation in *Ab* ATCC 17978 found no obvious link between KatA and ROS protection [25]. Still, the abundance pattern of ESGC$_{ACB}$-0624 indicate that this cluster may contribute to virulence in pathogenic members of the ACB clade. More comprehensive studies are needed to elucidate if and how ESGC$_{ACB}$-0624 is involved in the infection process.

## Metabolic adaptation–Micronutrient acquisition is refined in the ACB-clade

Essential metals, such as iron and zinc, are actively sequestered by the host to starve invading pathogens [81]. This likely results in a strong selective pressure for the pathogenic ACB clade to optimize scavenging systems such that the reduced bioavailability of these metal ions in the host can be counterbalanced. Acquisition systems for iron, whose limited availability at the host-pathogen interface is considered one of the key obstacles for invading and persisting within the human host, are a showcase example. The iron transporter system *feoABC* represents the evolutionary core of iron uptake. It is complemented, in many but not all taxa [33] both inside and outside of the ACB clade, by the baumannoferrin cluster (Fig 8). Two further clusters extend the spectrum of iron uptake systems exclusively in the ACB clade. ESGC$_{ACB}$-0498 represents the 2,3-dihydroxybenzoic acid synthesis cluster (*entAB*), which synthesizes a siderophore precursor [82]. ESGC$_{ACB}$-0368 and 0369 together resemble the acinetobactin biosynthesis clusters *bauA-F*, *basA-I* and *barAB*. A third cluster, ESGC$_{ACB}$-0485, that very likely

represents an ABC-type $Fe^{3+}$-hydroxamate transport system seems to extend the diversity of iron uptake systems in the ACB clade even further. It encodes a substrate-binding protein, an iron complex ABC transporter (permease), an ATP-binding protein, and an N-Acetyltransferase protein (GNAT family). The AraC-family-like transcriptional regulator, which is located downstream on the opposite strand in $ESGC_{ACB}$-0485, likely controls the expression of this cluster. In line with this operon-like organization, these genes are jointly downregulated under mucin-rich conditions [83]. The complex and seemingly redundant infrastructure for iron uptake in the ACB clade seems at odds with a recent study in *A. baumannii* ATCC 17978, which stated that acinetobactin is the only system that is necessary for *A. baumannii* to grow on host iron sources [84]. Here we show that this conclusion does not generalize to the entire ACB clade. The pathogens *A. nosocomialis* and *A. seifertii*, for example, lost the acinetobactin cluster (cf. Fig 5). It is conceivable that the diversity of iron acquisition systems is an adaptation to diverse niches each requiring different strategies of iron scavenging.

Given the essentiality of zinc (Zn), it is not surprising to see that also $Zn^{2+}$ uptake was refined on the lineage towards the ACB clade. The Zn uptake system Znu, including the distal *znuD* gene, which facilitates resistance to human calprotectin-mediated $Zn^{2+}$ sequestration [25], is evolutionarily old and part of the genus-wide core genome (Fig 8). The histidine utilization (Hut) system (*hutCDUHTIG*, $ESGC_{ACB}$-0215) is prevalent in the ACB clade, though not exclusively. This system ensures the bio-availability of $Zn^{2+}$ via the histidine catabolism both under high availability and starvation of $Zn^{2+}$. However, it requires histidine to be abundant. Interestingly, the most recent acquisition in Zn metabolism is the putative metallochaperone, ZigA. The corresponding gene resides directly adjacent to $ESGC_{ACB}$-0215, and thus is likely an evolutionary more recent extension of this cluster. *zigA* was found active only under Zn starvation, where it increases the bioavailability of Zn also under histidine depletion [85] and counteracts nutritional starvation.

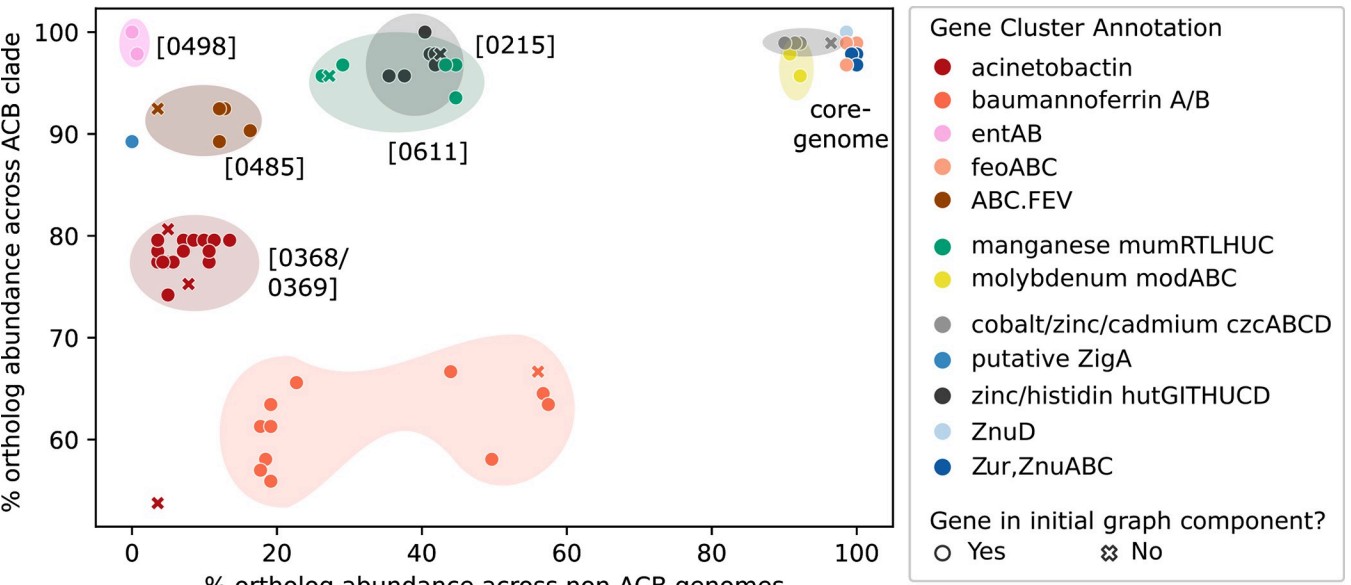

**Fig 8. Abundances of micronutrient acquisition genes/clusters of Ab ATCC 19606 within and outside the ACB clade.** Absolute abundances were based on cluster-evaluation using microsynteny. Gene clusters are annotated with the $ESGC_{ACB}$ identifiers in Fig 5. Genes that are part of a cluster but missed by the initial ESGC compilation share the same color but are marked as x. Gene clusters at the top right, e.g., feoABC, represent micronutrient acquisition clusters belonging to the genus' core genome. Gene clusters at the top left, e.g., entAB, are confined to the ACB clade where they are ubiquitously present.

Manganese ($Mn^{2+}$) is required only in small amounts and is mostly used for coping with reactive oxygen species (ROS), as $Mn^{2+}$, other than $Fe^{2+}$, does not promote the Fenton reaction that converts $H_2O_2$ to highly damaging hydroxyl radicals [86, 87]. Therefore, Mn uptake systems should be prevalent in bacteria frequently exposed to ROS stress, particularly in the pathogenic *Acinetobacter* strains. Thus far, only one Mn acquisition system, *mumRTLUHC* [88], has been identified in *Acinetobacter spp*. This system is represented by ESGC$_{ACB}$-0611 and plays an essential role in protecting *A. baumannii* against calprotectin-mediated Mn depletion by the host and contributes to bacterial fitness in a murine pneumonia model [88]. ESGC$_{ACB}$-0611 is found throughout the genus though less frequently in clades that are more distantly related to the ACB clade (cf. Figs 5 and 8). Within the HA-clade, several species including *A. tjernbergiae*, *A. junii*, *A. beijnerickii*, and *A. haemolyticus* lack both the putative manganese transporter gene *mumT* and the gene encoding a putative hydrolase *mumU* (S5 Fig id:0611). Within IC 3, several strains lack the entire cluster. These taxa either found alternatives to $Mn^{2+}$-dependent processes for coping with oxidative stress, are more vulnerable to ROS, or they scavenge $Mn^{2+}$ via a mechanism that is still hidden in functionally uncharacterized gene clusters.

In summary, we see a clear signal that the ACB clade is enriched for genes and gene clusters that functionally complement the genus-wide available and evolutionarily old metal uptake systems. In line with the reinforcement hypothesis, these more recently acquired clusters seem particularly important for metal scavenging during infection, i.e. when the metals are actively sequestered by the host [84,85].

## Carbohydrate metabolism—Evolution towards nutritional flexibility

The ability of individual *Acinetobacter* strains to utilize a broad spectrum of carbon sources is important for their adaptation to different environments, including the human host [45,89–92]. However, it is largely unknown when the corresponding metabolic pathways were acquired during *Acinetobacter* evolution, how widespread they are, and if and to what extent they are connected to the pathogenicity of the ACB clade. More than 20 of the shortlisted ESGCs$_{ACB}$ represent pathways that shuttle metabolites into the carbohydrate metabolism of the bacterium (Fig 5 in red font, and see Fig 9 for a selection), many of which are prevalent in the human body. The corresponding gene clusters mostly channel these metabolites into catabolic processes (see below). However, the genes involved in the glucose/gluconate metabolism seem to fuel anabolic processes.

## Glucose/gluconate metabolism

Glucose and gluconate serve as carbon and energy sources for few species in the genus *Acinetobacter*, e.g., *A. soli*, *A. apis*, and *A. baylyi*. For *A. baylyi* ADP1 it was shown that the glucose catabolism involves the Entner-Doudoroff pathway [93]. Members of the ACB clade have lost the ability to use glucose and gluconate as a carbon source [94] (see also S6 Fig and S7 Table). It is, thus, surprising that we find the genetic infrastructure to feed both molecules into the bacterial metabolism almost exclusively in the ACB clade.

ESGC$_{ACB}$-0112 comprises the gluconate permease (GntP) that shuttles gluconate from the periplasm into the bacterial cell (Fig 9, yellow pathway). The cluster further encodes the kinase GntK, which phosphorylates gluconate into 6-phosphogluconate, and the enzymes Edd and Eda of the Entner-Doudoroff pathway, which link to the pentose phosphate pathway that produces pyruvate. Members of the ACB clade also possess two variants of a glucose dehydrogenase (*gdh*), which catalyze the reaction from D-glucose to D-gluconate in the periplasm [95]. The membrane-bound variant (*gdhA*) forms together with an outer membrane porin a cluster

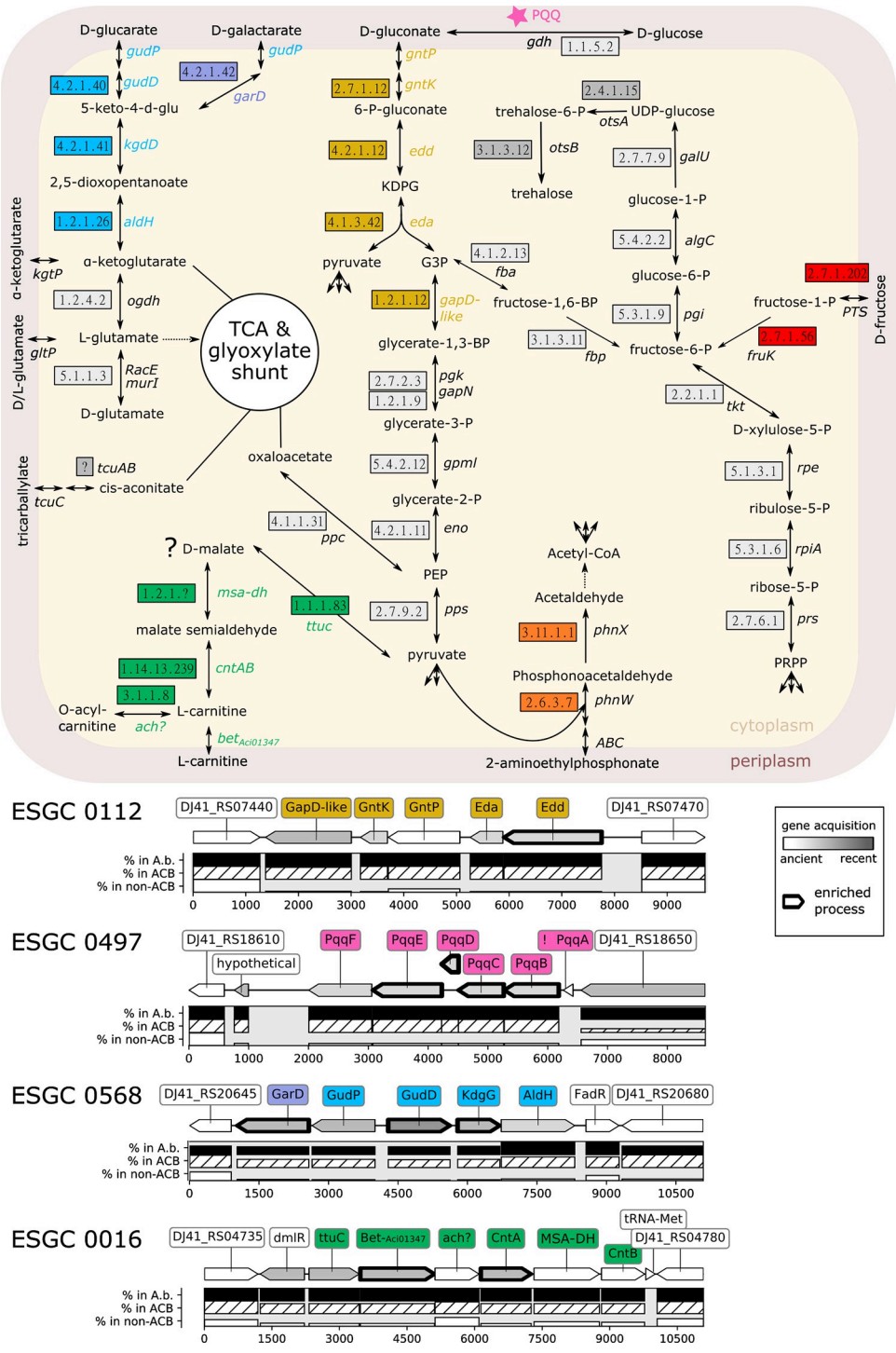

**Fig 9. Members of the ACB clade have extended their basal carbohydrate metabolism.** The pathway map shows a model integrating four metabolic pathways represented by the ESGC_ACB-0112 (Entner-Doudoroff pathway; yellow), 0497 (biosynthesis of Pyrroloquinoline quinone; magenta), 0568 (glucarate/galactarate catabolism; blue/violet) and 0016 (carnitine catabolism; green) into the bacterial carbohydrate metabolism (grey boxes). Abbreviations: KDPG— 2-keto-3-deoxy-6-phosphogluconate; G3P –glycerinealdehyde-3-phosphate. The corresponding gene clusters are shown below the pathway map with one protein-coding flanking gene on either side that is not part of the cluster. The layout follows Fig 7. ESGC_ACB-0112: GapD-like–glyceraldehyde-3-phosphate dehydrogenase [1.2.1.12]; GntK— gluconokinase [EC:2.7.1.12]; Edd—phosphogluconate dehydratase [EC:4.2.1.12]); Eda—2-dehydro-3-deoxyphosphogluconate aldolase / (4S)-4-hydroxy-2-oxoglutarate aldolase [EC:4.1.2.14 4.1.3.42]; GntP—gluconate

permease [E2.7.1.12]. Note, GntP shares high sequence similarity with a gluconate transporter (H+ symporter) in *Escherichia coli* (98% coverage, 45% identity). ESGC$_{ACB}$-0497: pqqA-F. No abundance profile is shown for pqqA, since its length excluded it from orthology prediction (indicated by an '!'. See main text for details). ESGC$_{ACB}$-0568: GarD—galactarate dehydratase [EC:4.2.1.42]; gudD—glucarate dehydratase [EC:4.2.1.40]; kdgD—5-dehydro-4-deoxyglucarate dehydratase [EC:4.2.1.41]; aldH—2,5-dioxopentanoate dehydrogenase [EC:1.2.1.26]; gudP—MFS transporter, D-glucarate/D-galactarate permease; FadR—Fatty acid metabolism regulator protein. ESGC$_{ACB}$-0016: ttuc—D-malate dehydrogenase [EC:1.1.1.83]; Bet-Aci01347—glycine/betaine transporter Aci01347; ach?–putative acylcarnitine hydrolase [EC:3.1.1.8]; CntAB–carnitine monooxygenase reductase subunit A and B [EC:1.14.13.239]; MSA-DH–malic semialdehyde dehydrogenase [EC:1.2.1.?]. Further colored pathways represent the clusters ESGC$_{ACB}$-0627 (2-aminoethylphosphonate metabolism, orange), ESGC$_{ACB}$ 0453 (fructose transport/metabolism, red), ESGC$_{ACB}$-0064 (trehalose biosynthesis, dark grey), and ESGC$_{ACB}$-0078 tricarballylate metabolism, brown). Their corresponding genomic regions are available in S5 Fig.

of two genes, ESGC$_{ACB}$-0287 (cf. S1 Data id:0287), which is ubiquitous across *Acinetobacter* spp. We note that the porin is orthologous to OprB in *P. aeruginosa*, where it facilitates the diffusion of various sugars—including glucose—into the periplasm. The second, soluble Gdh (*gdhB*) is confined to and nearly ubiquitous in the ACB clade (S7 Fig and S6 Table: id: HOG3408).

The prosthetic group for both Gdh, pyrroloquinoline quinone (PQQ), is a small, redox-active molecule that serves as a cofactor for several bacterial dehydrogenases. ESGC$_{ACB}$-0497 comprises six genes that together represent the PQQ biosynthesis pathway: *pqqABCDE* and an additional membrane-bound dipeptidase referred to as *pqqF* in *Klebsiella pneumoniae* [96] (cf. Fig 9, pink). All genes reside contiguously on the same strand suggesting an operonic structure. The complete cluster is present in almost all genomes of the ACB clade, although we had to manually confirm the presence of *pqqA*, because its length (40 amino acids) is below the length cutoff of the ortholog assignment tool. Only 31 out of 2436 strains in the ACB clade have lost the ability to synthesize PQQ, among them the model strain *A. baumannii* ATCC 17978. Cluster abundance outside the ACB clade is low (<20%), but it is present in all strains of species with demonstrated ability to assimilate glucose and gluconate (i.e. *A. soli*, *A. baylyi*, and *A. apis*, cf. [94]; S1 Data id:0497).

The holoenzymes *GdhA* and/or *GdhB*, in theory, could establish a gapless route for glucose via this gluconate 'shunt' into the cell for further degradation via the Entner-Doudoroff pathway, even in the absence of a dedicated Glucose transporter. Why then do none of the tested strains in the ACB clade grow with glucose as sole carbon and energy source? We hypothesize that they utilize this route for anabolic processes exclusively, e.g. for the production of polysaccharides as it was demonstrated for *P. aeruginosa* [97].

## Carbohydrate catabolism

The ACB clade has substantially increased its repertoire of catabolic pathways for alternative carbon sources compared to taxa outside this clade [94]. For a small number of mostly hand-picked *A. baumannii* strains, previous studies have experimentally confirmed the ability to grow on tricarballylate and putrescine, malonate, butanediol and acetoin, phenylacetate, muconate, glucarate, galactarate (mucate), and 4-hydroxyproline as sole carbon sources [45,89]. Our analyses identified the corresponding gene clusters among the ESGC$_{ACB}$. Hence, the ability to use these resources is prevalent in the ACB clade, whereas non-ACB species have to rely largely on different carbon sources. We will highlight two examples that likely represent an adaptation to humans as a host.

D-glucarate (saccharate) is a major organic acid in human serum [98]. ESGC$_{ACB}$-0568 comprises all necessary genes for glucarate and galactarate (mucic acid) degradation (Fig 9). In *Salmonella enterica* serovar Typhimurium deletion of the D-glucarate/D-galactarate permease

ortholog attenuated virulence [99]. Further, galactarate digestion was shown to increase the colonization fitness of intestinal pathogens in antibiotic-treated mice and to promote bacterial survival during stress [100]. ESGC$_{ACB}$-0568 is almost exclusively confined to the ACB clade. This may indicate that this cluster contributes to colonization and virulence in pathogenic *Acinetobacter* species. It is therefore interesting that within *A. baumannii* the cluster is almost absent in IC2 strains (0.08% prevalence in Set-F; *cf.* Fig 5).

Carnitine is essential for the oxidative catabolism of fatty acids in humans [101]. ESGC$_{ACB}$-0016 comprises six genes necessary for catabolizing carnitine [102]. A LysR-type transcriptional regulator likely controls the activity of this cluster. The remaining five genes represent a putative tartrate dehydrogenase (*ttuC*), a BCCT-family carnitine transporter (Aci01347), a generically annotated alpha/beta hydrolase which possibly catalyses the conversion of D-acylcarnitine into L-carnitine (see Fig 9, green), and the genes encoding the two subunits of the carnitine monooxygenase CntA and CntB. The latter two genes are separated by a gene that is tentatively annotated as an NAD-dependent succinate-semialdehyde dehydrogenase. However, two lines of evidence indicate that the precise function of this gene as well as that of the putative tartrate hydrogenase might both differ. In the literature, the putative succinate-semialdehyde dehydrogenase is speculated to act as malic semialdehyde dehydrogenase [102], an enzyme that converts malate semialdehyde into malate. Further, the putative tartrate dehydrogenase belongs to the KEGG orthologous group KO7246 which is annotated as a D-malate dehydrogenase. The product of the latter enzyme, pyruvate, can be further processed into oxaloacetate, which serves as a substrate for the tricarboxylic acid (TCA) cycle, or into acetyl CoA (cf. Fig 9). Assuming that the putative malic semialdehyde dehydrogenase produces D-malate, then this cluster should allow the members of the ACB clade to utilize D-malate as a carbon source if an appropriate transporter is present. We tested this hypothesis and confirmed that *Ab* ATCC 19606 grows on D-malate (S6 Fig), which corroborates initial growth experiments [103]. We note that the unusual production of the D-malate enantiomer rather than L-malate would have a further interesting implication. It potentially allows the bacterium to accumulate D-malate in conditions when carnitine is abundant, without interfering with the stoichiometry of the remaining substrates of the TCA cycle.

Thus far, two *A. baumannii* strains have been shown to use carnitine as sole carbon source [104]. We evaluated exemplarily that the absence of the cluster indeed correlates with *Acinetobacter* inability to grow on carnitine and is not functionally complemented by an alternative degradation pathway. Both, *A. baylyi* ADP1 and the *A. calcoaceticus* type strain (DSM 30006), which both lack the ESGC$_{ACB}$-0016, did not grow on carnitine after 24h (S6 Fig).

The clusters abundance profile reveals that the ability to metabolize carnitine occurs also outside the ACB clade (31 strains in Set-R; cf. Fig 5). However, 25 of these strains were isolated from infected patients, 2 from hospital sewage water, and only 3 strains were sampled from the environment (cf. S8 Table). The isolation origin of the remaining strain is unknown. The presence of the carnitine cluster, therefore, correlates surprisingly well with the pathogenic potential of a strain and it will be interesting to test a causal dependence. In support of causality, we find that the carnitine cluster is absent in *A. calcoaceticus*, the only species of the ACB clade that is nonpathogenic or at least has substantially reduced virulence (see S1 Text: Section ESGC$_{ACB}$-0016).

## Novel carbon sources

The functional annotations of the ESGC$_{ACB}$ in Fig 5 indicate that the list of potential carbon sources for *A. baumannii* and other members of the ACB clade is still incomplete. We find degradation pathways for xanthine, 2-aminoethylphosphonate, acyclic terpenes, vanillin/valinate, taurine, and anthranilate (see S1 Text: Section Gene clusters (ESGCs) not discussed in

main manuscript). To the best of our knowledge, none of these metabolites have been considered as potential carbon sources for *A. baumannii*, although a putative Xanthine dehydrogenase has been characterized previously via heterologous expression in *E. coli* [105]. One cluster is, however, particularly interesting because it intertwines the bacterial carbohydrate metabolism with the human tryptophan catabolism. ESGC<sub>ACB</sub>-0452 (Fig 10A) harbors the kynurenine

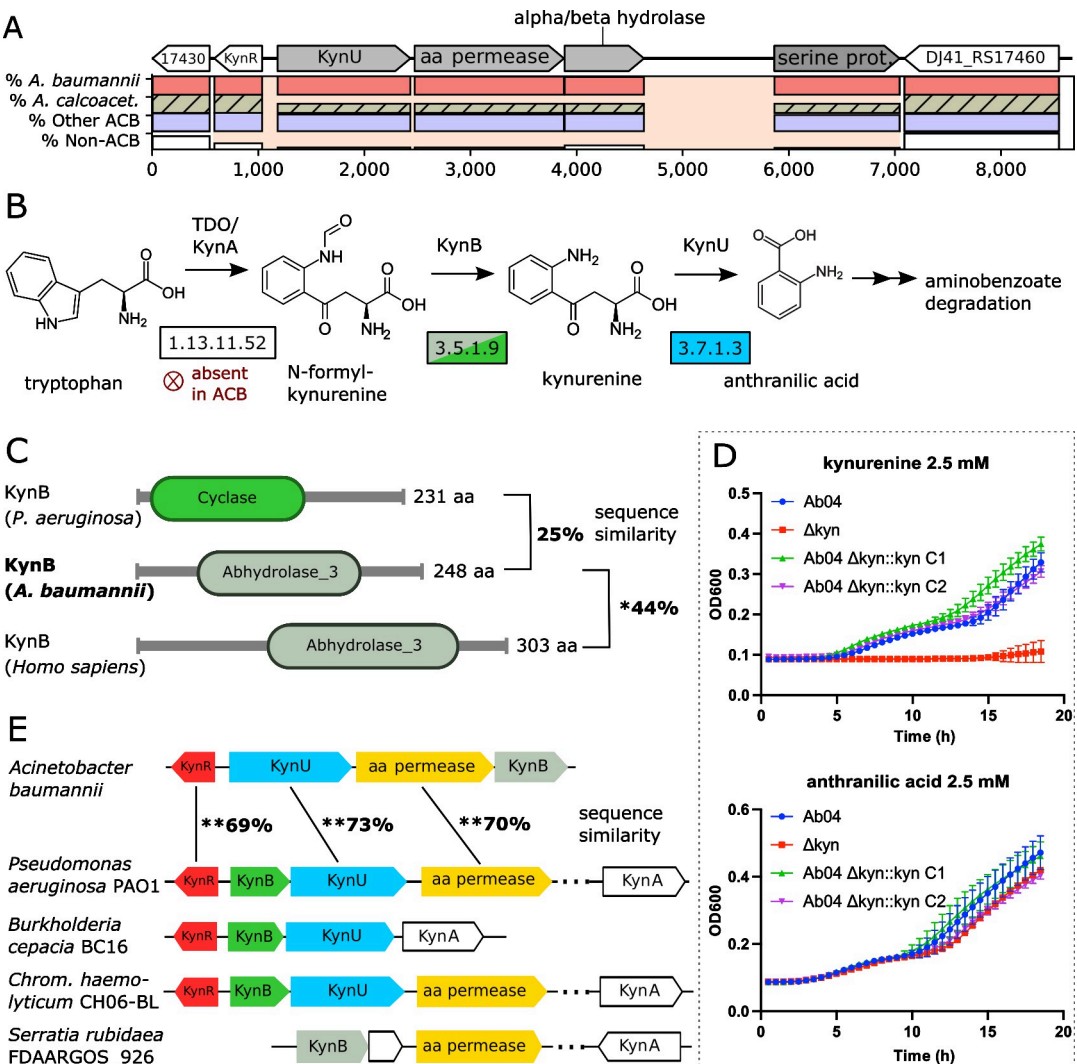

**Fig 10. Evolutionary and functional characterization of the A. baumannii kynurenine pathway.** (A) The locus of ESGC<sub>ACB</sub>-0452 in A. baumannii ATCC 19606. KynR—Lrp/AsnC family transcriptional regulator; KynU—kynurenine hydrolase; amino acid (aa) permease (uncharacterized); alpha/beta hydrolase (uncharacterized; putative kynB); serine proteinase. Cluster layout follows that of Fig 7. (B) The Kynurenine pathway of tryptophan degradation. EC numbers of the enzymes catalyzing the individual reactions are given in the boxes. KynB is represented in two versions in bacteria (see subfigure C), which are represented by different colors. (C) The alpha/beta hydrolase shares a significant sequence similarity and the presence of the Abhydrolase_3 Pfam domain (PF04199) with the human kynurenine formamidase (KynB or KFA) but not with KynB of P. aeruginosa, which instead harbors a Cyclase Pfam domain (PF04199) suggesting an independent evolutionary origin. (D) The kynurenine cluster is necessary and sufficient for growth on kynurenine but not on anthranilic acid. Growth of Ab04, Ab04 Δkyn, and of two Ab04 Δkyn::kyn strains (C1 and C2) on minimal medium supplemented with kynurenine (top) and anthranilate (bottom). (E) Phylogenetic profile of the A. baumannii Kyn cluster. Numbers between the corresponding genes in Ab ATCC 19606 and PA01 represent percent sequence similarity on the amino acid level. KynB of Ab ATCC 19606 and PAO1 are not significantly similar. The Kyn cluster is shared among many proteobacterial (opportunistic) pathogens where the Pseudomonas-type KynB is prevalent. The human-type KynB has homologs e.g., in Serratia (including S. rubideae). KynA is almost always encoded at a distant locus and entirely absent in Acinetobacter.

hydrolase KynU, which catalyzes the cleavage of kynurenine to anthranilic acid and alanine (Fig 10B). Within the same cluster, we identified an AsnC-type transcriptional regulator, a putative amino acid permease, and a gene generically annotated as an alpha/beta-hydrolase. Notably, this hydrolase is listed as an ortholog to the human kynurenine formamidase (KynB) in the OMA database [106] with which it shares the same domain architecture (Fig 10C). Together with its evolutionarily stable localization in the vicinity of KynU, this provides evidence that the *A. baumannii* hydrolase resembles a hitherto overlooked KynB. This enzyme transforms formyl-kynurenine into kynurenine and acts immediately upstream of KynU in the Kynurenine pathway of tryptophan degradation (*cf.* Fig 10B). A similar gene cluster was recently described in *P. aeruginosa* (Pae) [107,108], however with two notable exceptions: Kyn-$B_{Aba}$ is substantially more similar both in sequence and domain architecture to the human KynB than the $KynB_{Pae}$ (Fig 10C), and we found no trace of an enzyme that catalyzes the formation of N-formyl-kynurenine from tryptophan in *A. baumannii*.

We next confirmed that the presence of the kynurenine cluster allows *A. baumannii* to grow on kynurenine as a sole carbon- and energy source (Fig 10D). A deletion of the Kyn-cluster abolished growth on kynurenine but neither on anthranilate, the product of KynU (Fig 10D), nor on casamino acid or tryptophan (S8 Fig). In a last step, we investigated the phylogenetic profile of the Kyn-cluster in greater detail (Fig 10E). Within *Acinetobacter*, the cluster is almost exclusively present in the pathogens of the ACB clade, and it has been lost several times independently within *A. calcoaceticus*. Along the same lines, we find the Kyn-cluster in a proteobacteria-wide screen across more than 1,000 species (see S9 Table for the list of genomes and identifiers) only in a few taxa, of which many are opportunistic human pathogens (S2 Data). Taken together, our findings provide strong evidence that members of the ACB clade possess the genetic infrastructure to interfere with the tryptophan metabolism of humans, as it was already shown for *P. aeruginosa* [108]. This opens up a novel and hitherto unexplored route of how these pathogens interacts with its human host.

## Discussion

In contrast to more virulent bacteria, the opportunistic nosocomial pathogens from the genus *Acinetobacter* pursue a resist-and-persist strategy [27]. Instead of pinpointing individual key determinants of bacterial virulence, this requires the unraveling of a likely broad and less specific genetic basis conveying adaptation to clinical environments and, at the same time, to the human host. Experimental studies, either *in vitro* bringing the advantage of controlled experimental conditions, or *in vivo* with the advantage of a realistic infection model have provided fundamental insights into the pathobiology of *Acinetobacter* [21,22,45]. However, pathogens encounter diverse environments during host infection [109]. The resulting selective landscape is complex, and therefore hard to reproduce in an experimental setup. Virulence determinants that are relevant only under specific conditions or whose functions blend in with the bacterial metabolism are easy to miss. Evolutionary approaches can close this gap. They focus on the signal if a gene or a gene cluster likely contributes to pathogenicity, reflected by their prevalence preferentially in pathogens, independent of the precise conditions when it is active. Here, we have charted the genetic specifics of the pathogenic ACB clade at a resolution ranging from a genus-wide overview down to individual clonal lineages within *A. baumannii*. To ease the future integration of evolutionary evidence with functional studies, we have developed the *Acinetobacter* dashboard Aci-Dash. This is the first web application that allows the community to mine the abundance profiles of genes encoded in 232 representative genomes together with their functional annotations and their connection to virulence factors.

Comparative genomics studies across the genus *Acinetobacter* have been performed before (e.g. [11,110]). The integration of orthology assignments with shared synteny analyses at a scale that spans several thousand *Acinetobacter* genomes is yet unprecedented. The ESGC$_{ACB}$ detected here form relevant starting points for further unravelling the regulatory and functional network in a human pathogen. They result in a rich set of testable hypotheses whose experimental validations will likely deepen the understanding of the genetic basis of *Acinetobacter* pathogenicity. Moreover, the lineage specific absence of ESGC within the pathogenic clade, most prominently demonstrated by the loss of the QS-NRPS cluster together with the Csu cluster in the IC8, or of several clusters in *A. calcoaceticus*, helps to better predict in which characteristics individual strains, clonal lineages, and species differ from the prototype of an *Acinetobacter* pathogen.

The functions conveyed by these clusters are diverse and many imply a role in *Acinetobacter* pathogenicity: Quorum sensing and biofilm formation [97], ROS response [111], and micro-nutrient acquisition [81]. On top of these, the abundance of gene clusters involved the carbohydrate metabolism, a largely uncharted area of *Acinetobacter* virulence factors [34], indicates that members of the ACB clade follow a general evolutionary trend towards greater metabolic flexibility, which is common to many bacterial pathogens [37,109]. Such generalists have a selective advantage over niche-specialists in environments that are frequently disturbed or altered [112].

Metabolic interaction with the host, in particular those involving amino acids, have an interesting further implication, that has thus far not been considered in the context of *Acinetobacter* pathogens. They can modulate regulatory systems involved, for example, in the finetuning of the host immune response [37,113]. The Kynurenine (Kyn) pathway detected in this study is a likely example of this connection. This cluster is almost entirely confined to the pathogenic members of the ACB-clade. Interestingly, and in contrast to the human pathogen *P. aeruginosa*, which also harbors a Kyn pathway, the key enzyme, IDO (KynA), that allows the bacteria to directly metabolize tryptophan via the kynurenine pathway is missing (cf. Fig 10B). How then do they fuel this pathway? Interestingly, tryptophan depletion via the kynurenine pathway is an important human immune defense mechanism upon bacterial infection [114]. Here, we provide first time evidence that pathogenic *Acinetobacter* species from the ACB clade can use the intermediate metabolites of the host response as additional carbon- and energy sources, likely further promoting its growth. However, the ability to degrade the intermediates of the human tryptophan catabolism has a further interesting implication for the host-pathogen interaction. While it was shown that *P. aeruginosa* produces elevated levels of kynurenine to inhibit ROS production and aid bacterial survival [115], *Acinetobacter* species in possession of the Kyn cluster must pursue a different strategy. It is tempting to speculate that they can interfere with the homeostasis of the human immune system, in particular with its suppressive effect on T cells and Natural killer cells, by scavenging kynurenine from their environment. The now uncontrolled production of ROS could cause excessive host tissue damage [80, 83]. Though at a high cost, this would allow the bacteria to use the rich nutrient resources enclosed in the host cells.

In summary, antimicrobial resistance is one of largest threats to global health. On the example of *Acinetobacter*, we have shown that the incorporation of a broad evolutionary perspective can pinpoint individual genes or entire pathways that result in novel and viable hypotheses of how the bacteria persist, feed off and interact with the host. At an early phase of drug development, these candidates provide promising anchor points from which the development of new therapeutic strategies to either prevent or treat *Acinetobacter* infections can be initiated.

## Methods

### Data acquisition

The full data set (Set-F) comprises all assemblies in the NCBI RefSeq data base (version 87) stating 'Acinetobacter' in the 'organism' field (S1 Table). From Set-F, we selected in total 232 representative strains covering all available type, reference and representative genomes, as well as all validly named species (https://apps.szu.cz/anemec/Classification.pdf) for which a genome sequence was available at the study onset are represented. We further picked genomes of several *A. baumannii* strains that are of interest due to e.g., their context of isolation as well as representatives of eight international clones. Lastly, we included genomes from Set-F into Set-R that allow for an increase of the fraction of total phylogenetic diversity covered without compromising the quality (cf. S1 Text: Section Taxon Set Construction). An overview of Set-R is provided in S2 Table.

To compile the *Proteobacteria* data set (n = 1363), we selected all *Proteobacteria* represented in NCBI RefSeq data base (version 204) and selected one representative per species, which was either annotated as "reference" or "representative" strain.

### International Clones (IC) and MLST assignments

For all strains in Set-F, we determined the sequence type with MLSTcheck v2.1.17 [116] using two different schemes, Oxford [117] and Pasteur [118], that were obtained from the PubMLST website (http://pubmlst.org/abaumannii/). All members of the *A. baumannii* clade were assigned to an international clone (IC) whenever we found literature evidence that the predicted sequence types and IC were unambiguously linked using the following publications as a source: [39,118–127]. The final assignments are provided in S1 and S2 Tables based on strain typing results provided in S10 Table.

### Phylogenetic diversity

Phylogenetic diversity scores of SET-F and SET-R were computed with PDA v.1.0.3 [128] using the options *-k = 234:3027* and *-if = handpicked.list* based on the ML tree of SET-F. 25 low quality assemblies were pruned prior to this analysis to avoid overestimation because of long branches resulting from sequencing errors. A detailed description of the quality assessment is provided in S1 Text: Section Taxon Set Construction.

### Average nucleotide identity

All genome sequences within Set-R were pair-wise aligned with Nucmer v3 [129], and the Average Nucleotide Identities (ANIm) were calculated with the script average_nucleotide_identity.py from the Python package pyani v0.2.7 [130] using the following options: *-i./genomes -o./output/ -m ANIm -g—gmethod seaborn–maxmatch*.

### Ortholog search

All against all orthology searches were performed with OMA standalone v.2.2.0 [48] and default parameter settings, except for decreasing the minimum length threshold for sequences considered (MinLen) to 40 residues. Targeted ortholog searches were performed with fDOG [131] using the OMA orthologous groups from SET-R to train the profile hidden Markov models.

## Inference of the pan and core gene sets (SET-R)

The pan-genome size was calculated as the sum of the number of OGs and number of strain specific proteins (i.e. proteins without orthologs). For the rarefaction analysis, we identified the number of new orthologs and singletons per added genome in Set-R. The core genome was defined as the subset of OGs where each taxon of Set-R contributed exactly one ortholog. However, the strict definition of the core genome, yielded very small core genome estimates due to qualitative differences and incomplete draft genomes (S1 Text: Section Core-genome reconstruction). We therefore relaxed the core genome definition and allowed a core-gene to be absent in 1% of the genomes (max. 3 out of the 232) in Set-R (see S1 Text for further details). To obtain these values as a function of the number of genomes considered, we simulated a sequential inclusion of genomes in SET-R. Following the approach from Tettelin et al. [132], Pan- and core genome sizes were extrapolated by fitting the power law function $yP = \kappa Pn^{\gamma} + c$ and an exponential decaying function $yc = \kappa c \exp[-n^{*}\tau c] + \Omega$, respectively, with nonlinear least-squares (Levenberg–Marquardt algorithm). Given the large taxon set in our study, we limited the simulation to 100 random permutations of a sequential inclusion.

## Phylogenetic tree reconstruction

SET-R: Multiple sequence alignments (MSAs) for each orthologous group were generated with MAFFT-LiNSI [133] (v7.394, default parameters). Next, each protein's CDS was obtained and PAL2NAL v14 [134] with the option *-codontable 11* was employed to infer protein sequence guided nucleotide MSAs. Supermatrices built from concatenated MSAs served as input for the maximum likelihood (ML) tree reconstruction. The best fitting substitution model (GTR + empirical base frequencies + 10 substitution rate categories) was determined using IQ-Tree v1.6.8 [135] using the option *-m TEST*, and the following parameters were used for the ML tree reconstruction with IQ-Tree: *-m GTR+F+R10 -nt 6 AUTO -bb 1000 -alrt 1000*. Additionally, statistical branch supports were assessed with 1000 repetitions of UF bootstraps and SH-aLR branch tests. Trees were outgroup-rooted with *Moraxella catarrhalis* (strains BBH18 and FDAARGOS_213) and *Alkanindiges illinoisensis* DSM 15370.

SET-F: The phylogenetic tree reconstruction of SET-F followed the same general work flow as described for Set-R. However, to decrease the computational burden, trees were computed on the amino acid sequence alignments. MAFFT was run with the '--auto' parameter. Supermatrices of resulting protein MSAs were used as input for ML tree reconstruction with IQ-TREE (-alrt 1000 -bb 1000 -nt 8 -m LG+I+G+F). Majority-rule consensus trees were computed with SplitsTree v4.14.4 [136].

## Inference of hierarchical orthologous groups and reconstruction of last common ancestor dispensable and core gene sets

Hierarchical orthologous groups (HOGs) for the Set-R were inferred from the pairwise OMA orthology relations and the consensus ML species tree using the GETHOG algorithm [137] as implemented in OMA stand-alone v. 2.2.0. The Set-R pan genome was stratified by assigning each HOG to the internal node of the tree that represents the last common ancestor (LCA) of the two most distantly related taxa in the HOG. Clade-specific losses of a gene were inferred when all members of a clade lacked an ortholog that was assigned to an evolutionarily older node. On this basis, we reconstructed the pan-genomes for each internal node of the tree as following: We united all genes assigned to internal nodes on the path from the root to the node under study and removed the union of genes that have been lost on this path.

## GO annotation and GO term enrichment analysis

All 502,095 unique protein accessions represented in SET-R were mapped to uniprotKB identifiers (UniProt accessed 9[th] of Febr. 2018) to obtain the annotated gene ontology (GO) terms [138]. Significantly enriched GO terms were identified using a two-tailed Fischer's exact test at a significance level of 0.05. Multiple test correction was done by computing the false discovery rate for each term and considering only terms with an FDR < 0.05. For GO term enrichment analyses at the individual internal nodes of the tree, we used the LCA pan gene sets as background sets (population) and the set of genes assigned to this node as test (study) set. Both sets were limited to include only genes for which orthologs are represented in at least 50% of the taxa descending from this node (analysis without this cutoff is included in S5 Table). Information about the GO terms were pulled from http://purl.obolibrary.org/obo/go/go-basic.obo (accessed 22[nd] of June 2018). Visualization of tree maps were performed with REVIGO [139].

## Prediction of secreted proteins

Unless the subcellular localization of a protein was provided by uniprot, we predicted its subcellular localization with Psortb v3.06 [140] and ngLOC v1.0 [141]. All proteins classified as 'Extracellular' or 'OuterMembrane' by either tool were combined into a set of secreted and accessible proteins. Conflicting predictions or cases where Psortb labeled the localization of a protein as 'unknown' were resolved in favor of the ngLOC classification as it demonstrated higher precision for these classes when we benchmarked both against a test-set of experimentally verified proteins published by Shen et al. [142].

## Annotation of protein function

For each protein, we considered, where available, its functional annotation provided in RefSeq, in the uniprot database, and the assigned GO terms. Additionally, we annotated the proteins with KEGG Orthology (KO) identifiers [50] using GhostKoala v2.2 [143]. Eventually, for proteins with reciprocal best blast hit orthologs in *A. baumannii* ATCC 17978, we transferred the functional annotation provided by the COG database [49] (accessed in May 21[st] 2020).

## Virulence factor identification

HOGs representing known virulence factors were identified via blastp searches (v2.10.1) [144] against a custom database of virulence factors. To compile this database, we united entries of PATRIC [23] and VFDB [51] (both accessed September 23[rd] 2020). Subsequently, we clustered the proteins at 95% sequence similarity with cd-hit v4.6.4 [145] using the options *-G -al 0.95 -s 0.9 -c 0.95* to reduce redundancy. A HOG represents a virulence factor if any of the subsumed orthologs has a hit with >50% alignment coverage and an e-value <0.01. For all such instance the best hit's annotation was transferred.

## Identification of Evolutionarily Stable Gene Clusters (ESGC)

Using the genome of *A. baumannii* ATCC 19606 as a reference, we identified clusters of consecutive genes with highly similar phyletic profiles. For this purpose, each protein-coding gene's corresponding HOG and its profile were analyzed to obtain an 8-dimensional feature vector comprised of the following values: (i-vi) clade-specific fractions of total taxa in *A. baumannii* (B), *A. calcoaceticus* (CA), *A. haemolyticus* (HA), *A. baylyi* (BA), *A. lwoffii* (LW), *A. brisouii* (BR), and *A. qingfengensis* (QI), (vii) the fraction of the total phylogenetic diversity of the ACB clade covered, and (viii) the label of the inner node the HOG was assigned to (cf. Fig 3). Four genes were excluded from the orthology search due to their short length. For these, we

imputed the values using the mean values of the two flanking genes. Based on these vector representations, we computed a pairwise dissimilarity matrix, using the Gower dissimilarity index [146]. We then arranged the genes in a graph, where a vertex between a gene and its downstream neighbor was drawn, if a) their pairwise dissimilarities is smaller than the $5^{th}$ percentile of the gene's dissimilarity distribution across the full gene set or b) if the condition *a* is met by the two genes flanking the gene. The resulting set of connected components were then extracted as candidate evolutionary units.

## Candidate ESGC abundance statistics

For each gene in a candidate ESGC, we computed its retention difference (RD) as the difference in the fraction of taxa within the ACB clade subtracted by the fraction of taxa outside the ACB clade harboring an ortholog. The candidate ESGCs were then ranked by the median of the RD across all genes in the cluster. As a further measure, we devised the cluster-conservation score difference (CCD), which is calculated similar to the RD score, but this time assessing presence of the cluster rather than that of an individual gene. A cluster was considered present only if at least 80% of its genes (orthologs) were identified and at least 25% of the gene order was conserved in the genome. Here, we treated the cluster as a set of ordered and oriented (considering direction of transcription) two-element tuples. Clusters with a CCD below 0.25 were not further considered. For each of the resulting top 150 ranked clusters, we inspected cluster conservation across the taxa in Set-R and Set-F using Vicinator v0.32 (https://github.com/BIONF/Vicinator). Cluster boundaries of an ESGC were manually curated, when the Vicinator analysis indicated a miss due to an individually bloated abundance profile (false-positive orthologs or paralogs).

## Kynurenine (Kyn) cluster deletion mutant and growth experiments with *A. baumannii* Ab04

Ab04 mutant with deletion of the locus of ESGC$_{ACB}$-0452 (Ab04 Δkyn) were constructed as described previously [147]. Briefly, a FRT site-flanked apramycin resistance cassette was amplified from a variant of pKD4 [148] with primers comprising 18–25 nucleotides matching in sequence the flanking regions of the locus of ESGC$_{ACB}$-0452 (see S11 Table). Also, upstream and downstream regions of ESGC$_{ACB}$-0452 were amplified and the obtained fragments were assembled by overlap extension PCR. The PCR product was electroporated into *A. baumannii* Ab04 electrocompetent cells carrying pAT04, which expresses the RecAB recombinase induced with 2mM IPTG [1]. Mutants were selected with apramycin and integration of the resistance marker was verified by PCR. To remove the resistance cassette, electrocompetent mutants were transformed with pAT03 plasmid (which expresses the FLP recombinase) and apramycin-sensitive clones of unmarked deletion mutants were obtained. Finally, the mutant strains were confirmed by antibiotic resistance profile, PCR and genome sequencing.

To generate genetic complementation, the genes deleted in the Ab04 Δkyn mutant strain were cloned into the pUC18T-mini-Tn7T-Apr vector and introduced to the mutant strain via four-parental mating methods as described previously [149]. Briefly, overnight cultures from the recipient strain, HB101(pRK2013) strain, EC100D(pTNS2) strain, and *E. coli* containing the pUC18T-mini-Tn7T-Apr construct were normalized and mixed 1:1. The suspension was centrifuged, re-suspended in 25 ml of LB, spotted on a pre-warmed LB agar plate and incubated overnight at 37˚C. The bacteria were scraped from the plate, resuspended in 1 ml of LB, and plated on LB agar plates supplemented with chloramphenicol and Apramycin to select transconjugants. Correct insertion of the constructs was verified by PCR amplification and sequencing using the primers listed in S11 Table.

Ab04 WT, Ab04 Δkyn mutant strains, and Ab04 Δkyn::kyn were grown in lysogeny broth (LB) liquid medium under shaking conditions (200 rpm) at 37˚C. Overnight cultures were washed three times with PBS and diluted to an $OD_{600}$ of 0.01 in 150 μl of M9 minimal medium (1X M9 salts (Becton Dickinson, cat # 248510); 2mM MgSO4; 0.1 mM CaCl2) supplemented with 0.2% casamino acids (M9CAA), 2.5 mM L-kynurenine (Sigma, A8625) or 2.5 mM anthranilic acid (Sigma, A89855) in 96-well plates, followed by incubation at 37˚C under shaking conditions in a BioTek microplate spectrophotometer. The $OD_{600}$ values were measured every 30 min for 18h. Three independent experiments were performed with three wells per assay for each strain and condition.

## Growth experiments for Carnitine, Malate, and Glucarate

*A. baumannii* ATCC 19606, *A. baylyi* ADP1, and *A. calcoaceticus* DSM 30006 strains were grown at 37˚C (*A. baumannii*) or 30˚C (*A. baylyi* and *A. calcoaceticus*) in mineral medium (MM) that consists of 50 mM phosphate buffer, pH 6.8, and different salts (1 g $NH_4Cl$, 580 mg $MgSO_4 \times 7 H_2O$, 100 mg $KNO_3$, 67 mg $CaCl_2 \times 2 H_2O$, 2 mg $(NH_4)_6Mo_7O_{24} \times 4 H_2O$, 1 ml SL9 (per liter: 12.8 g Titriplex, 2 g $FeSO_4 \times 7 H_2O$, 190 mg $CoCl_2 \times 6 H_2O$, 122 mg $MnCl_2 \times 4 H_2O$, 70 mg $ZnCl_2$, 36 mg $MoNa_2O_4 \times 2 H_2O$, 24 mg $NiCl_2 \times 6 H_2O$, 6 mg $H_3BO_3$, 2 mg $CuCl_2 \times H_2O$ per l medium; pH 6.5) [150] and 20 mM of the given carbon source. Precultures were grown in MM with 20 mM Na-acetate as carbon source. Each value is the mean of +/- S. E.M. of at least three independent measurements. Growth curves were fitted manually.

## Supporting information

**S1 Text. Supplementary Text.** Contains supplemental sections covering additional information on the taxon set compilation, a statistical exploration of all protein-coding genes, genomes and orthologs in Set-R, details on the method and workflow of the ESGC prediction, and provides additional results and discussions for the predicted $ESGCs_{ACB}$ including several clusters not discussed in the main manuscript.
(DOCX)

**S1 Fig. In Set-R there is a significant difference in the number of coding sequences between the ACB clade members and non-members.** (A) Comparison of the number of coding sequences (CDS) per genome between members and non-members of ACB clade across *Acinetobacter*. It reveals a significant difference. ACB clade members, on average, contain 14% more protein coding genes. (B) Correlation matrix for a range of summary statistics on genome level across SET-R. Colored cells indicate value of spearman correlation coefficient [−1,1]. The descriptive statistics analyzed are explained in S2 Table. (C) Phylogenetic diversity of the orthologous groups (OGs) calculated from the sum of branch lengths of the subtree spanned by the taxa represented in an OG. This distribution contrasts the taxa belonging to the ACB clade vs. the total phylogenetic diversity. Data points are colored black if the corresponding OG belongs to the set of genus-wide core genes that were also used for phylogeny reconstruction. OGs represented closer to the upper left corner are especially interesting as they are approaching ubiquitous presence within the ACB clade but are rare in the rest of the genus (colored orange for illustrative purposes).
(PDF)

**S2 Fig. Majority-rule consensus phylogeny of 232 Acinetobacter strains represented in SET-R.** A high resolution image of the majority-rule consensus dendrogram of the Set-R taxa as shown in Fig 2A. Branches supported by only two out of three partition trees are indicated with dashed lines, branches supported by only one partition are not resolved. Leaf labels

colored in green indicate changed species assignments. Such changes can either (i) correct (i.e. the original species assignment was at odds with the species assignment based on phylogenetic and ANI evidences), (ii) newly specify (i.e. the original species assignment was set to "unknown" (sp.)) or (iii) de-specify (the original species assignment could not be confirmed by phylogenetic evidences, and no alternative assignment was possible. The species label was set to "sp.") the species assignments as retrieved from NCBI RefSeq at the time of download.
(PDF)

**S3 Fig. Maximum likelihood (ML) tree for Set-F based on partition 1 of the core gene set (296 proteins).** Branch labels denote percent bootstrap support. The newick strings for the ML trees from all three partitions are given in S3 Data. A high-resolution figure of the consensus tree is provided in S10 Fig.
(PDF)

**S4 Fig. Hierarchically clustered ANIm heatmap across Set-R combined with phylogenetic information reveals unknown species diversity within the ACB clade.** The figure shows the color-graded average nucleotide identity (ANIm, all vs all) of the genomes across SET-R (both axes). Strains with high genomic identity generate clusters of high sequence identity (>95%: increasingly saturated red; 95%: white, <95%: saturating blue) along the diagonal. These clusters and cutoffs are typically used for bacterial species delineation. In the ACB clade, we observe a large and distinct cluster with 98–100% sequence identity (rounded) for the *A. baumannii* species (top left). Following the diagonal to the bottom right corner, the strains of *A. seifertii* (2, n = 2) and *A. nosocomialis* (3, n = 5) are clustered similarly. The percent identities decrease for the following two clusters of genomes with mainly *A. calcoaceticus* species assignments (4 and 5). Specifically, only three pairs in cluster 4 and one pair in cluster 5 reach the species threshold ANI. Next to putatively misidentified *A. calcoaceticus*, cluster 5 features a strain of the tentative species *A. oleivorans* (strain CIP 110421) as well as the unassigned *A.* sp. WC-141 and *A.* sp. NIPH 817 suggesting the existence of undescribed species in the ACB clade. Two further genomes, here located between cluster 5 and cluster 6 are putatively mislabeled strains *A. baumannii* strain 573719 and *A. pittii* ANC 4050. These strains again potentially represent undescribed species in the ACB clade. Clusters 6 and 7 comprise the strains from *A. lactucae* and from *A. pittii*, respectively. S12 Table contains the full matrix of pairwise ANIm in a tabulated format.
(PNG)

**S5 Fig. Genomic regions of all ESGC along the genome of ATCC 19606.** Graphical representations of the genomic regions for each ESGC with RD > 0 (see Methods) along the genome of ATCC 19606 with abbreviated abundance profiles and functional annotations.
(PDF)

**S6 Fig. Growth of *A. baumannii* ATCC 19606, *A. calcoaceticus* DSM 30006 and *A. baylyi* on different carbon sources.** *A. baumannii* ATCC 19606 (△), *A. calcoaceticus* (□) and *A. baylyi* ADP1 (○) were grown in mineral medium with 20 mM D-malate (A), gluconate (B) or carnitine (C) as carbon source. Each value is the mean of +/- S. E.M. of at least three independent measurements.
(PDF)

**S7 Fig. Ortholog abundance profiles and functional characterizations of *gdhA* and *gdhB*.**
(PDF)

**S8 Fig. Growth of *A. baumannii* Ab04, Ab04 Δkyn, and of Ab04 Δkyn:kyn on casamino acid and tryptophane.**
(PDF)

**S9 Fig. Numbers of HOG innovations and losses at each node of the majority-rule consensus tree of Set-R.** A consensus tree representation of Set-R with each inner node of the tree annotated with the number of HOGs associated to it as well as the number of lost HOGs in the subsumed clade according to the rules of Dollo (+) Parsimony (see Methods). All HOGs are provided in S13 Table. The nodes are labeled with an incremental id. On the lineage of *A. baumannii* we used the following replacements in the manuscript: NODE_1 = "ACB+BR", NODE_8 = "ACB+LW", NODE_9 = "ACB+BA", NODE_10 = "ACB+HA", NODE_11 = "ACB", NODE_12 = "BNS", NODE_13 = "B". Tip labels are represented as NCBI RefSeq Identifiers. Tip labels also show unique assembly accession and, if applicable either clone type assignment or (corrected) species assignment in form of the first four letters of the species names.
(PDF)

**S10 Fig. Majority-rule consensus tree (dendrogram) for Set-F summarizing the information in the maximum likelihood trees based on the three partitions of the core gene set.**
(PDF)

**S11 Fig. Treemaps of enrichment biological processes as generated by REVIGO (*cf*. S5 Table).**
(PDF)

**S1 Data. Phylogenetic profiles and microsynteny plots in html format for the curated top 150 ESGCs$_{ACB}$ across Set-F as produced by the Vicinator tool.**
(TAR.GZ)

**S2 Data. Phylogenetic profiles and microsynteny plots in html format for ESGC$_{ACB}$-0452 and the kynurenine pathway cluster of *P. aeruginosa* across a sample of all *Proteobacteria*.**
(TAR.GZ)

**S3 Data. ML trees for all three partitions in newick format including support values.**
(TAR.GZ)

**S1 Table. Full list of genomes used in this study.** The set of assemblies represented in Set-F is provided in tabular format as extracted from NCBI RefSeq. Among other, the table lists RefSeq assembly and sample accessions, taxonomy ids (on strain and species level if available), species labels, and strain labels, assembly status, submission date, submission institute, and ftp link to resource.
(XLSX)

**S2 Table. Representative genomes selected with corrections and descriptive statistics.** The subset of assemblies in Set-R is provided in tabular format. The table lists RefSeq assembly and sample accessions, taxonomy ids, species labels, and corrected labels (see main text), strain names, clone type assignments (via MLST and literature) sampling site, and sampling year (where available) of the strains. The table further features the results of our genome sequence-specific quantitative and qualitative analyses. The descriptive statistics are described in the documents' sheet 'ColumnLegends'.
(XLSX)

**S3 Table. Orthology matrix as calculated from the all vs. all orthology search across Set-R (OMA orthologous groups).** (1) The column header contains the labels of the NCBI RefSeq assembly accession of each genome. Row indices refer to the orthologous group identifier. The matrix fields contain the NCBI RefSeq protein sequence accession if an ortholog was identified and left empty otherwise. (2) List of orthologous groups identifiers constituting the core

genome by our definition. These were used for tree reconstruction.
(XLSX)

**S4 Table. Phylogeny-based taxonomic classifications of the genomes in Set-F into the defined clades.** This table comprises 15 sheets. Each sheet lists the NCBI RefSeq assembly accession and taxon label (species and strain label) of the genomes that were phylogenetically classified to belong to the following clades: B clade (*A. baumannii*), S clade (*A. seifertii*), NO clade (*A. nosocomialis*), BNS clade (three aforementioned), L clade (*A. lactucae*), PI clade (*A. pittii*), CA clade (*A. calcoaceticus*), CPL clade (three aforementioned), ACB clade, HA clade, BA clade, LW clade, BR clade, QI clade, and the outgroup. Genome assignments to *A. baumannii* which were at odds with its phylogenetic placement in our study are indicated by a blue line. In the same way, we highlighted assignments to *A. pittii* in light orange and *A. calcoaceticus* in dark orange.
(XLSX)

**S5 Table. Results of consecutive GO term enrichment analysis.** Sheet 1 shows the significantly enriched GO terms ($p_{adjusted} < 0.001$) for proteins in those HOGs at a node where an ortholog was detected in at least 50% of the subsumed taxa. If genes of the same HOG are the source for multiple enriched terms, only the most specific term (highest depth) was kept for the final results. The raw results of the GO term enrichment analysis are presented in Sheet 2. The results of the GO term enrichment analyses considering all HOGs at a node are provided in Sheet 3. Detailed explanations of the column headers are placed in the sheet 'ColumnLegends'. The test results displayed in Sheet 3 were extracted to generate input tables for REVIGO (sheets 4–8). The resulting treemaps are shown in S11 Fig.
(XLSX)

**S6 Table. ESGC identification and underlying data.** This file gives detailed information about the ESGC identification and provides the underlying data. Sheet 1 gives for each gene along the genome of *A. baumannii* ATCC 19606 the number of taxa per clade (and per international clone type) that harbor an ortholog. Sheet 2 gives, for each gene, the input vector used for the dissimilarity calculations, the individual thresholds (5$^{th}$ percentile) and, as an example, the calculated dissimilarity between each gene and the gene immediately upstream. The full matrix containing all pairwise dissimilarity calculations for the prediction of clusters is deposited as txt format on figshare (https://doi.org/10.6084/m9.figshare.16910974.v1). Sheet 3 lists the identified graph components along with abundance statistics (median of the proteins in a component) across the clades (both absolute and relative), retention difference between ACB vs. non-ACB (RD), and CCD scores. Sheet 4 lists detailed contextual information for the components including all functional annotations from various sources. Detailed explanations of the column headers for all tables are placed in the sheet 'ColumnLegends'.
(XLSX)

**S7 Table. Summary statistics of collected experimental evidence found in the literature for the assimilation of >100 potential carbon sources for *A. baumannii*.**
(XLSX)

**S8 Table. List of all non-ACB genomes across Set-R that harbor ESGC$_{ACB}$-0016 including meta information regarding sampling and isolation.**
(PDF)

**S9 Table. Assembly accessions and species/strain labels for the *Proteobacteria* sample (n = 1363).**
(XLSX)

**S10 Table. In silico MLST classifications of the genomes in Set-F.** Sheet 1 displays the results of the classification using the Oxford-scheme and Sheet 2 using the Pasteur-scheme. Sheets 3–10 list NCBI RefSeq assembly accession and strain labels for all genomes in the *A. baumannii* clade where the MLST classification allowed a mapping to one of the 8 international clones. (XLSX)

**S11 Table. List of primers used for Ab04 mutant deletions Δkyn mutant strains.** (XLSX)

**S12 Table. Raw values of the pairwise ANIm comparisons shown in S4 Fig.** (XLSX)

**S13 Table. Hierarchical orthology matrix as calculated from the all vs. all orthology search across Set-R along the phylogenetic tree (OMA hierarchical orthologous groups, HOGs).** Sheet 1: The column header contains the labels of the NCBI RefSeq assembly accession of each genome. The second column (LCA) indicates the associated inner node of the HOG in the reconstructed phylogenetic tree. The values either refer to the node label on the lineage of *A. baumannii* or, if the node is not on the lineage, an incremental node id. The corresponding nodes together with summary statistics are provided in S9 Fig. Row indices refer to the hierarchical orthologous group ID. The matrix fields contain the NCBI RefSeq protein sequence accession if an ortholog was identified otherwise they are left empty. Sheet 2 lists the identifiers of HOGs with prevalence in at least 231 genomes. These were used for the definition of core components during cluster identification. (XLSX)

**S14 Table. Results of BLAST search for homologues of the QS cluster against non-*Acinetobacter* genomes.** (XLSX)

## Acknowledgments

The authors wish to thank all researchers for making annotated genome sequences available to the public domain, Prof. Alexandr Nemec for providing on his website an accessible and up-to-date resource of the current species nomenclature of *Acinetobacter*, Vinh Tran for support with fDOG, and Ruben Iruegas for helpful discussion.

## Author Contributions

**Conceptualization:** Bardya Djahanschiri, Ingo Ebersberger.

**Data curation:** Bardya Djahanschiri.

**Formal analysis:** Bardya Djahanschiri.

**Funding acquisition:** Alexander Goesmann, Stephan Göttig, Gottfried Wilharm, Mario F. Feldman, Ingo Ebersberger.

**Investigation:** Bardya Djahanschiri, Gisela Di Venanzio, Jesus S. Distel, Jennifer Breisch, Stephan Göttig, Gottfried Wilharm, Ingo Ebersberger.

**Methodology:** Bardya Djahanschiri, Beate Averhoff, Ingo Ebersberger.

**Project administration:** Ingo Ebersberger.

**Resources:** Bardya Djahanschiri, Gisela Di Venanzio, Alexander Goesmann, Mario F. Feldman, Ingo Ebersberger.

**Software:** Bardya Djahanschiri, Marius Alfred Dieckmann, Alexander Goesmann.

**Supervision:** Alexander Goesmann, Beate Averhoff, Mario F. Feldman, Ingo Ebersberger.

**Validation:** Bardya Djahanschiri, Mario F. Feldman, Ingo Ebersberger.

**Visualization:** Bardya Djahanschiri, Ingo Ebersberger.

**Writing – original draft:** Bardya Djahanschiri, Ingo Ebersberger.

**Writing – review & editing:** Bardya Djahanschiri, Gisela Di Venanzio, Jesus S. Distel, Jennifer Breisch, Beate Averhoff, Stephan Göttig, Gottfried Wilharm, Mario F. Feldman, Ingo Ebersberger.

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
