## [Decision Letter · Decision Letter 0]

1 Mar 2022

Dear Dr Ebersberger,

Thank you very much for submitting your Research Article entitled 'Evolutionarily stable gene clusters shed light on the common grounds of pathogenicity in the Acinetobacter calcoaceticus-baumannii complex' to PLOS Genetics.

The manuscript was fully evaluated at the editorial level and by independent peer reviewers. The reviewers appreciated the attention to an important topic but identified some concerns that we ask you address in a revised manuscript

We therefore ask you to modify the manuscript according to the review recommendations. Your revisions should address the specific points made by each reviewer.

[LINK]

Yours sincerely,

Xavier Didelot

Associate Editor

PLOS Genetics

Josep Casadesús

Section Editor: Prokaryotic Genetics

PLOS Genetics

Reviewer's Responses to Questions

**Comments to the Authors:**

Reviewer #1: The manuscript by Djahanschiri et al. submitted to PLoS Genetics provides a comprehensive overview of the evolution of stable gene clusters across the Acinetobacter genus and in particular their relationship with the ACB complex and subsequently the virulence capabilities of pathogenic species. The rationale, the approach and the outcomes have been well-described. Further, the balance between bioinformatics and placing them in appropriate biological context was reached across the entire manuscripts which is highly appreciated. Immediately, this renders the manuscript and the online aci-dash tool a valuable resource in the field of Acinetobacter biology.

General comments:

A phenotypic conformation was achieved for one of their findings, kynurine utilisation, but it seems rather preliminary. Although this may have been adequate if it was performed for multiple findings across the manuscript, such as QS, I believe further controls for kynurine utilisation are appropriate, as detailed below.

Consider dampening the conclusions regarding associations between gene presence/absence and a role in virulence. The results are indicative of these roles.

The numbering of the supplementary material is confusing.

Please don’t insert too many sentence breaks, such as seen in line 37, it restricts readability.

Specific Comments:

Line 29: The functional context (eg GO) was largely the means of study in this manuscript too.

Line 32: no italics for atcc 19606

Line 43: Improve clarity on “show”, eg bioinformatic vs phenotypic.

Line 47: Instead consider emphasising that this study represents a very comprehensive resource for future research into novel therapeutic strategies.

Line 54: Dampen the word “essential”

Line 58-59: Remove “comprising the A. baumannii-calcoaceticus complex”

Line 65-66: this reads clunky, please revise.

Line 73: Delete “with”

Line 77: Revise this sentence.

Line 83: Please include additional info on international clones, consider providing inforamtion regarding IC8 in particular.

Line 92: The entire section on the three main approaches does not read very well. Please make this more concise and don’t include the numerical list.

Line 118: What is the difference between “genome plasticity” and "flexible genome" as described in line 115. This is a bit confusing.

Line 125: The balance between background and intro to new findings is inadequate.

Line 135- 138: Be mindful of repetition between the abstract and this section.

Line 140-145: Probably too much detail and this has now transitioned to background again.

S1 Table: I can only find 38 A. nosocomials, whilst it indicates 69 in Figure 5, please clarify.

Fig 1: This figure serves no primary function in this manuscript. Please move to sup material.

Line 209: Using both However and though in one sentence does not aid readability.

Line 213: Please include a ref to 2C when covering environment/host presence.

Line 217-221: Please revise wording

Line 226: Remove “at least”

Line 241-242: Please refer to figure/table and/or name these.

Line 332: Why are CsuC and D conserved and not lost?

Line 332: This does not appear true for the “L” and “CA” clades, please verify.

Line 359: This intro seems a bit lengthy, consider moving this material to the background section.

Line 390: Please use abbreviations if introduced earlier.

Line 392: Include full stop after (Fig 8)

Line 401: Why is the charge included here but not for Zn?

Line 416: Or inhibit niches that minimize ROS exposure?

Line 434: Individual = specific/various?

Line 466: Should this be “do”?

Line 551: Controls such as complementation and growth in the presence of tryptophan are needed. Further, does competition with more readily available carbon sources influence the activity of Kyn?

Line 826: Remove “chosen”

Reviewer #2: Djahanschiri B, et al, by including more than 3,000 Acinetobacter genomes surveyed the phylogenetic framework integrating orthology-based phylogenetic profiling and microsynteny conservation.

The functional integration of the subsumed genes achieved by ESGCs analyses show metabolic pathways, transcriptional regulators residing next to their targets but also tie together sub-clusters with distinct functions to form higher-order functional modules. Moreover, Kynurenine (Kyn) cluster deletion mutant and growth experiments with A. baumannii Ab04 as well as growth experiments supplemented with selected sugars have been studied.

The topic is interesting and the subject is according to the scope of the Journal. The manuscript is well written and the study will increase our understanding of how a highly critical pathogen interacts with its host (genetic point of view). It definitely deserves to be published and is a valuable contribution to the PlOS Genetics in terms of high importance to researchers in the field as well as high importance and broad interest to researchers in genetics and genomics. Indeed, I have provided few remarks/questions on the text below:

General comments.

Several studies highlighted that A. baumannii is able to express several virulence factors, including those required for biofilm formation, desiccation resistance, secretion systems, micronutrient acquisition systems, and twitching motility, as well as adherence to and invasion of human epithelial cells in vitro and in vivo (I draw you attention to some as follows: https://doi.org/10.1128/mSystems.00604-20;
https://doi.org/10.1016/j.ijid.2020.09.352;
https://doi.org/10.1038/ncomms13414;
https://doi.org/10.1038/s41598-018-21841-9;
https://doi.org/10.1128/mBio.02193-16).

As highlighted by the authors, genomic analyses could critical pathogen interacts with its host, which substantially eases the identification of novel targets for innovative therapeutic strategies. It should be noted that he capacity of an organism to adapt via changing its spatial, structural or functional relation to its external or internal milieu. On the other hand, each environmental conditions could define the behavior of a pathogen and its adaptation.

As concluded by the authors their genomic analyses resulted in the hypotheses on how pathogenic Acinetobacter interact with and ultimately infect their human host. This conclusion is too ambitious without direct in vitro/in vivo investigations on AB strains upon contact with their host cells and several influential environmental conditions such as pH, osmotic stress, OMPs expression, receptor signaling pathways either during normal growth or upon in vitro and in vivo. Accordingly, several specific downstream signaling pathways are involved that enable A. baumannii to enter alveolar host cell epithelia that could be totally display differently from their genomic content, gene assembly and even genetically closed strains. These above-mentioned points should be somehow mention in this study.

Minor comments:

Line 37 and Abstract: the following “They unveil, at an unprecedented resolution, the genetic makeup that coincides with the manifestation of the pathogenic phenotype in the last common ancestor of the ACB clade” is not clear. Please rephrase it in a simpler and understandable way.

Line 644: eight international clone types were included in this study. How and which parameters have been chosen to include them in the study. Are they selected for example, based on clonal diversity, phylogenetic index, etc?!

Experiments: Kynurenine (Kyn) cluster deletion mutant and growth experiments with A. baumannii Ab04 and growth experiments for carnitine, malate, and glucarate:

Why pre-cultures were grown in MM with 20 mM Na-acetate? And what is the reson to choose Carnitine, Malate, and Glucarate as carbon source). Only because of their sugar KEGG pathway??

What is the intention to select A. baumannii, A. baylyi and A. calcoaceticus strains? Please provide strains identity.

Are there any differences between M9 minimal medium and mineral medium (MM)!!

Where did you show the “Growth Experiments for Carnitine, Malate, and Glucarate”? these 2 simple experiments con not achieve following “they potentially interfere with the kynurenine-dependent fine-tuning of the human innate immune response” conclusion. Additional in vitro and in vivo experiments are required.

**Have all data underlying the figures and results presented in the manuscript been provided?**

Reviewer #1: Yes

Reviewer #2: Yes

PLOS authors have the option to publish the peer review history of their article (what does this mean?). If published, this will include your full peer review and any attached files.

Reviewer #1: No

Reviewer #2: **Yes: **meysam sarshar

---

## [Editor Report · Decision Letter 1]

4 Apr 2022

Dear Dr Ebersberger,

We are pleased to inform you that your manuscript entitled "Evolutionarily stable gene clusters shed light on the common grounds of pathogenicity in the Acinetobacter calcoaceticus-baumannii complex" has been editorially accepted for publication in PLOS Genetics. Congratulations!

Yours sincerely,

Xavier Didelot

Associate Editor

PLOS Genetics

Josep Casadesús

Section Editor: Prokaryotic Genetics

PLOS Genetics

Comments from the reviewers (if applicable):

**Data Deposition**

http://datadryad.org/submit?journalID=pgenetics&manu=PGENETICS-D-22-00021R1

**Press Queries**

---

## [Editor Report · Acceptance letter]

12 May 2022

PGENETICS-D-22-00021R1 

Evolutionarily stable gene clusters shed light on the common grounds of pathogenicity in the Acinetobacter calcoaceticus-baumannii complex 

Dear Dr Ebersberger, 

We are pleased to inform you that your manuscript entitled "Evolutionarily stable gene clusters shed light on the common grounds of pathogenicity in the Acinetobacter calcoaceticus-baumannii complex" has been formally accepted for publication in PLOS Genetics! Your manuscript is now with our production department and you will be notified of the publication date in due course.

With kind regards,

Anita Estes

PLOS Genetics

On behalf of:
